
# Scale dependence of cirrus heterogeneity effects.
# Part I: MODIS thermal infrared channels

Thomas Fauchez[1,2], Steven Platnick[2], Kerry Meyer[3,2], Céline Cornet[4], Frédéric Szczap[5], and Tamás Várnai[6,2]

[1]Universities Space Research Association (USRA), Columbia, MD, USA
[2]NASA Goddard Space Flight Center, Greenbelt, MD, USA
[3]Goddard Earth Sciences Technology and Research, Universities Space Research Association, Columbia, MD, USA
[4]Laboratoire d'Optique Atmosphèrique, UMR 8518, Université Lille 1, Villeneuve d'Ascq, France
[5]Laboratoire de Météorologie Physique, UMR 6016, Université Blaise Pascal, Clermont Ferrand, France
[6]University of Maryland Baltimore County: Joint Center for Earth Systems Technology and the Department of Physics , Baltimore, MD, USA

*Correspondence to:* Thomas Fauchez (thomas.j.fauchez@nasa.gov)

**Abstract.** This paper presents a study on the impact of cirrus cloud heterogeneities on MODIS simulated thermal infrared (TIR) brightness temperatures (BT) at the top of the atmosphere (TOA) as a function of spatial resolution from 50 m to 10 km. A realistic 3-D cirrus field is generated by the 3DCLOUD model, and 3-D thermal infrared radiative transfer (RT) is simulated with the 3DMCPOL code. According to previous studies, differences between 3-D BT computed from a heterogenous pixel and 1-D RT computed from a homogeneous pixel are considered dependent, at nadir, on two effects: *(i)* the optical thickness horizontal heterogeneity leading to the homogeneous plane parallel bias (PPHB) and the *(ii)* horizontal radiative transport (HRT) leading to the independent pixel approximation error (IPAE). A unique but realistic cirrus case is simulated and, as expected, the PPHB impacts mainly the low spatial resolution results (above $\sim 250\ m$) with averaged values up to 5 - 7 K while the IPAE impacts mainly the high spatial resolution results (below $\sim 250\ m$) with average values up to 1 - 2 K. A sensitivity study has been performed in order to extend these results to various cirrus optical thicknesses and heterogeneities by sampling the cirrus in several ranges of parameters. For four optical thickness classes and four optical heterogeneity classes, we have found that, for nadir observations, the spatial resolution where the combination of PPHB and HRT effects is the smallest, falls between 100 m and 250 m. These spatial resolutions appear thus to be the best choice to retrieve cirrus optical properties with the smallest cloud heterogeneity-related total bias in the thermal infrared. For off-nadir observations, the average total effect is increased and the minimum is shifted to coarser spatial resolutions.

## 1 Introduction

In the context of global climate change, the representation and role of clouds are still uncertain. Cirrus clouds cover between 15 % and 40 % of the Earth's surface (Sassen et al. (2008)) and play an important role in Earth's the climate and radiative budget (Liou (1986)). The temperature difference between the cirrus cloud top and the Earth's surface leads to a warming of the atmosphere by cirrus clouds capturing a part of the infrared radiation emitted by the atmosphere and surface. Also,



cirrus clouds reflect part of the incident solar radiation into space due, but this albedo effect is generally negligible for high thin clouds. Thus, on average, cirrus clouds lead to a positive radiative effect (e.g. a greenhouse effect) except for cirrus with large optical thicknesses (greater than 10, Choi and Ho (2006)) or at low altitudes (below 8 km in the tropics, Corti and Peter (2009)). The radiative impact and evolution of cirrus clouds depends on numerous factors such as cloud altitude, cloud optical

and geometrical thickness, crystal shape and effective size. Consequently, we need to improve our knowledge by taking accurate observations of their optical properties.

Global satellite observations are well suited to follow and better understand cloud evolution and characteristics. Therefore, many satellites are dedicated to their observations from microwave to visible ranges. Cirrus optical thickness (COT) and ice crystal effective diameter (CED) can be retrieved from radiometric measurements using dedicated operational algorithms.

Many of these operational algorithms are developed for solar-reflectance channels, like that of the Moderate Resolution Imaging Spectroradiometer (MODIS), for the MOD06 product (Platnick et al. (2003); Yang et al. (2007)) or the Clouds and the Earth's Radiant Energy System (CERES) product (Minnis et al. (2011)) or of the Visible Infrared Imager Radiometer Suite (VIIRS, Platnick and et al. (2013)). Thermal infraRed (TIR) channels are currently used in the MOD06 dataset to retrieve cloud temperature/pressure/altitude (and in other datasets to retrieve ozone concentration and clear-sky temperature/moisture infor-

mation). However, several studies (Cooper et al. (2007); Cooper and Garrett (2010); Wang et al. (2011)) have shown that the TIR is better suited to retrieve cirrus COT and CED than visible and near infrared (VNIR) techniques such as the Nakajima and King method (NK, Nakajima and King (1990)), as long as the cirrus are optically thin enough (with a visible optical thickness between roughly 0.5 and 3) with CED smaller than $40 \ \mu m$. For example the Split Window Technique (Inoue, 1985) applied to the Advanced Very High Resolution Radiometer (AVHRR Parol et al. (1991)) and the Imaging Infrared Radiometer (IIR)

onboard CALIPSO ((Garnier et al., 2012, 2013)) is used to retrieve CED and COT from the brightness temperature difference of two different window channels. Based on the same spectral information, an optimal estimation method (OEM, Rodgers (2000)) is used for the Atmospheric Infrared Sounder V6 (AIRS, Kahn et al. (2014, 2015)) and in the research-level code of Wang et al. (2016b, a) for MODIS. Another advantage of the TIR is that measurements can also be obtained in nighttime conditions, which gives a distinct benefit compared to solar-reflectance channels for developing ice cloud climatologies. However,

in both VNIR and TIR optical property retrieval methods, each pixel is considered independent of its neighbors (independent pixel approximation, IPA Cahalan et al. (1994)) and fully homogeneous (homogeneous plane parallel approximation (PPHB Cahalan et al. (1994)). Theses approximations are mostly due to time constraints on 3-D forward radiative calculations, the lack of observation of the 3-D structure of the cloud, etc..

Many studies have been conducted in the solar spectral range to better understand the impact of cloud heterogeneities on cloud products. These studies primarily concern warm clouds such as stratocumulus (Varnai and Marshak (2001); Zinner and Mayer (2006); Kato and Marshak (2009); Zinner et al. (2010); Zhang and Platnick (2011); Zhang et al. (2012)) and show that the sign and amplitude of retrieval errors depend on numerous factors, such as the spatial resolution, wavelength, geometry of observation and cloud morphology. In the TIR and for ice clouds, Hogan and Kew (2005) show that radiative transfer (RT)

calculations using IPA can change the mean Top Of Atmosphere (TOA) radiative fluxes by $45 \ W.m^{-2}$ in the shortwave and by





15 $W.m^{-2}$ in the longwave. Chen and Liou (2006) show that the difference in the broadband thermal cooling rates is around 10 % when 3-D RT is compared to 1-D RT. Concerning IR radiances or Brightness Temperatures (BT), Fauchez et al. (2012, 2014) show that heterogeneity effects can significantly influence cirrus optical property retrievals at the 1 km scale of IIR thermal infrared observations, with potentially more than +10 K on TOA BT for heterogeneous pixels, depending also on the

cloud altitude. Fauchez et al. (2015) also show that these TOA BT effects result in an overestimate of the retrieved effective diameter by more than 50 % for small crystals (under 20 $\mu m$) and underestimate the retrieved optical thickness by up to 25 %. These errors could significantly influence the cirrus feedbacks assumed in global atmospheric models.

The impact of cloud horizontal heterogeneity depends on the spatial resolution of the instrument (pixel size) and the cloud type. For example, Davis et al. (1997) show for stratocumulus clouds that the heterogeneity impact on cloud optical thickness

retrieved from nadir visible radiometric measurements is at a minimum around a pixel size of 1 or 2 km for a small solar zenith angle (22.5°). Higher spatial resolutions enhance the IPA error (IPAE), which increases when the Fictive Light Particle (FLIP, Pujol (2015)) mean path (before absorption or cloud escape) is equal to or larger than the spatial resolution. Lower spatial resolutions have larger errors due to the homogeneous and plane parallel cloud assumption bias (PPHB), which increases when the assumed-homogeneous pixel size is increased. These results are very relevant as they can allow us to estimate the average

error due to cloud heterogeneity based on the spatial resolution of any space-borne radiometer. In addition, studies such as Davis et al. (1997) that attempt to identify the spatial resolution at which the error is at a minimum can help to define the ideal spatial resolution for future instruments. However, because such studies focus only on stratocumulus clouds, which are very different from cirrus and furthermore they are only valid for visible wavelengths, their conclusions cannot be simply extrapolated. In this paper, we focus our attention on cirrus clouds by simulating MODIS nadir TIR measurements (at the 8.52 $\mu m$, 11.01 $\mu m$,

12.03 $\mu m$ and 13.36 $\mu m$ wavelength, respectively), and compare the impact of horizontal cloud heterogeneity as a function of spatial resolution from 50 m to 10 km.

In Section 2, we present the 3DCLOUD model (Szczap et al. (2014)) used to simulate a realistic cirrus case study and the ice crystal optical property model used in MOD06, as well as the 3DMCPOL radiative transfer code (Cornet et al. (2010), Fauchez et al. (2012, 2014)), used to simulate the 3-D RT inside the atmosphere in the thermal infrared. In section 3 we describe the

heterogeneity and 3-D effects that impact the TOA BT at nadir for our cirrus cloud. In Section 4 we study the impacts of cirrus heterogeneities on TOA brightness temperatures viewed from nadir as a function of the spatial resolution for the above four MODIS TIR channels. The influence of the geometry of observations is discussed in Section 5. Summary and conclusions are given in Section 6.





## 2 Simulation of a realistic cirrus cloud field case study

### 2.1 Generation of a 3-Dimensional cirrus field

#### 2.1.1 Scale invariant properties

In order to study the impact of spatial resolution on cirrus heterogeneity effects, it is important to simulate as accurately as
possible the cloud inhomogeneity at the observational scale. Microphysical quantities such as liquid water content (LWC) or
ice water content (IWC), optical quantities such as extinction coefficient or radiative field such as radiances, reflectances and
brightness temperatures are not randomly distributed from small to large scales but often follow a power law in Fourier space
(Benassi et al. (2004); Cahalan and Snider (1989); Davis et al. (1994); Davis et al. (1996, 1997); Fauchez et al. (2014), etc.).
Indeed, Kolmogorov theory (Kolmogorov (1941)) shows that in the inertial domain, where the turbulence is isotropic and at
the equilibrium with large scales, spectral energy as a function of the wave number $k$ is described by a power law spectrum
$E(k)$ with an exponent $\beta \sim -5/3$ named spectral slope. We commonly say the $E(k)$ has scale invariant properties as expressed
by the following equation:

$$E(k) \propto k^{-\beta} \tag{1}$$

For scales smaller than the inertial domain, viscosity phenomena smooth and homogenize the fluid movement and the spec-
tral energy is no longer correlated with the wavenumber (Benassi et al. (2004)). The limit is not clearly defined because of
limitations due to instrument resolution. But theoretically, it could be defined as the scale of molecular dissipation, from a few
millimeters or more, depending on the turbulence intensity. The upper limit is defined as the scale where the spectrum becomes
flat (uncorrelated fluctuations). This scale can vary from one cloudy field to another. From in situ LWC airborne measurements,
Davis et al. (1994) and Davis et al. (1996) have estimated that the horizontal LWC spectral slope has a constant exponent of
about -5/3 between a few meters and a few tens of kilometers for three different stratocumulus clouds. Wood and Taylor (2001)
reached roughly the same conclusions for stratocumulus LWP. The situation is more complex for radiative quantities where
3-D effects (radiative smoothing and roughening) can modify the spectral slope (Oreopoulos and Cahalan (2005)). For example
Cahalan and Snider (1989) have shown that in satellite measurements (particularly from the TM radiometer on LANDSAT),
the spectral energy E(k) of radiances at the TOA follow a spectral law with a -5/3 exponent from the scale of 500 m to about
500 km; for scales less than 500 m, the spectral slope decreases to values close to -3 (Davis et al. (1997)).
Concerning cirrus clouds, Hogan and Kew (2005) showed using RADAR reflectivity that the IWC spectral slope exponent $\beta$
is equal to about -5/3 at the top of the cirrus from the scale of 1 meter to 100 km. But, they have also shown that the spec-
tral slope can decrease to -3 deeper in the cirrus if its geometrical thickness is very large (4 km in their case) and the cirrus
old enough (strong sedimentation process). Wang and Sassen (2008) have also shown, for one specific cirrus case, that the
spectral slope is close to -5/3 for small scales ( 500m-5km) but shows a -3 spectral slope for larger scales (5 km to 100 km).
This value is explained by the authors as the consequence of different dynamic processes such as vertical wind shear, thermal





stratification, and sedimentation processes and by the potentially uncommon cirrus structure. Using data from the CIRCLE-II airborne campaign, Fauchez et al. (2014) show that the horizontal spectral distribution of IWC and optical thickness follow a power law with $\beta \sim -5/3$ on the whole domain size (20 km). They also found the same power law at every cirrus altitude levels in the 532 nm backscattering coefficient measured by the Cloud-Aerosol Lidar with Orthogonal Polarization (CALIOP; Winker et al. (2009)). To summarize, except for particularly strong dynamical (mesoscale) processes such as those shown in Wang and Sassen (2008), the spectral slope of the horizontal distribution of IWC is typically -5/3. In this study, the size of our domain is small ($10 \times 10$km) and mesoscale processes can thus be neglected. We therefore assume, for our simulations, that the horizontal distribution of IWC follows a power law with a -5/3 exponent at every cloud level from the smallest cloud generator scale (50 m) to the domain scale (10 km) as show in Figure 1.

### 2.1.2 The cloud generator 3DCLOUD

To generate 3-D cloud structures, 3DCLOUD (Szczap et al. (2014)) first assimilates meteorological profiles (humidity, pressure, temperature and wind velocity) and then solves simplified basic atmospheric equations. Finally, a Fourier filtering method is used to constrain the scale invariant properties (by imposing the horizontal 2-dimensional (2D) distribution of IWC to follow a power law with -5/3 exponent at every cloud level), and to set the mean value and the heterogeneity parameter of these 3-D cloud structures. The heterogeneity parameter of optical thickness has been defined by Szczap et al. (2000) as $\rho_\tau = \sigma_\tau / \bar{\tau}$ with $\sigma_\tau$ the standard deviation of the optical thickness estimated for a particular pixel spatial resolution and $\bar{\tau}$ the averaged value of the optical thickness over the domain. The heterogeneity parameter is estimated without taking into account the holes in the cloud which are already related to the fractional cover parameter (here set to 1). Fu et al. (2000), Smith and DelGenio (2001), Buschmann et al. (2002), Carlin et al. (2002) and Hogan and Illingworth (2003) have shown using in situ or radiometric measurements that the heterogeneity parameter $\rho_\tau$ is typically between 0.1 and 1.5.

In Fig. 2 we can see the vertical profiles of the wind speed, temperature, relative humidity and ice mixing ratio assimilated by 3DCLOUD. These profiles are based on a mid-latitude summer meteorological profile modified to generate cirrus clouds (see for example Szczap et al. (2014)).

Figure 3 shows the optical thickness over the domain (a) and the 2-D IWC along the diagonal (red line in (a)) (b) generated using the meteorological profiles of Fig. 2 and by adjusting the optical thickness mean value while holding constant the -5/3 spectral slope of the IWC power spectrum. For the cirrus used in this study, the mean optical thickness is $\tau = 1.4$ at 12.03 $\mu m$, and the heterogeneity parameter of the optical thickness is $\rho_\tau = 1.0$. These values are consistent with those observed for cirrus clouds as shown in Table 1, which summarizes key cirrus properties listed in the literature with the range of possible values, the mean value and the value of the simulated cirrus for each parameter. The simulated cirrus field is thus suitable to study the impact of cloud heterogeneity on radiative transfer at various scales. Note that this paper is focused only on horizontal heterogeneities: we assume that the vertical variability of the geometrical and optical thickness is negligible compared to the





horizontal variability (see Fauchez et al. (2014, 2015)).

## 2.2 Optical properties

In this study, we use the same cirrus optical property parametrization as in the MOD06 product (Holz et al. (2015),Platnick
et al. (2016)), namely the severely roughened aggregate of solid columns parametrization of Yang et al. (2013). Note that TIR
retrieval techniques are often limited to effective diameters between 5 and 50 $\mu m$. The selection of this particle type instead
of another habit (or mixture of habits) is based on the study of Holz et al. (2015), who found that this habit provided better
consistency between the IR split-window technique and visible and near-/shortwave-/midwave-infrared (VNIR/SWIR/MWIR)
techniques, as well as with lidar retrievals. We assume a constant crystal effective diameter of 20 $\mu m$ throughout the cirrus
cloud. Note that TIR retrieval techniques are often limited to effective diameters between 5 and 50 $\mu m$. The choice of a crystal
effective diameter of 20 $\mu m$ falls thus almost in the middle of this range. The optical properties of this ice particle as a function
at each MODIS channel are shown Table 2 .

## 2.3 Radiative transfer

Radiative transfer computations are performed with the 3-D Monte Carlo code, 3DMCPOL (Cornet et al. (2010), Fauchez et al.
(2012, 2014)). In 3DMCPOL, the atmosphere is divided into 3-D volumes named voxels, with constant horizontal sizes (dx,
dy) and a variable vertical size (dz) that depends on the atmospheric and cloud vertical stratification. Inside the cloud, each
voxel is described by the cloud bulk scattering properties: the extinction coefficient $\sigma_e$, the single scattering albedo $\varpi_0$, the
phase function of the ice crystals and the temperature $T$.
3DMCPOL uses the local estimate method (LEM; Marchuk et al. (1980); Marshak and Davis (2005); Mayer (2009)), which
computes the contribution of emission, scattering or reflection events into the detector direction, attenuated by the medium
optical thickness between the place of interaction and the detector (Fauchez et al. (2014)). Atmospheric gaseous absorption is
parameterized using a correlated k-distribution (Lacis and Oinas (1991); Kratz (1995)) method combined with the equivalence
theorem (Partain et al. (2000); Emde et al. (2011)). The equivalence theorem is used in attaching a vector containing the at-
mospheric absorption attenuation to the FLIP, with the vector dimension being equal to the number of bins in the correlated
k-distribution. This allows for considerable savings in computational time.

In this study, we performed RT calculations for MODIS channels 29 (8.52 $\mu m$), 31 (11.01$\mu m$), 32 (12.03 $\mu m$) and 33
(13.63$\mu m$) in the TIR range. 3DMCPOL computes directly the radiances which are then converted into brightness temperatures
(BT), the quantity more commonly used in thermal infrared applications. Figure 4 shows the result of a 3-D BT computation at
12.03 $\mu m$ wavelength and 50 m horizontal spatial resolution for the "cirrus 1" scene. For this single wavelength and spatial res-
olution, 100 billion FLIPs are computed in 10 days on the NASA NCCS DISCOVER supercomputer (see acknowledgments)
for an accuracy of 0.5 K. As we will explained, RT computations are performed for the different thermal infrared channels for
1-D and 3-D, and for different spatial resolutions. This yields a large number of cases and a significant computational burden.





For this reason, and because Fauchez et al. (2014) showed that radiative heterogeneity effects are linked, to the first order to the optical thickness heterogeneity regardless how the optical thickness is distributed, we chose to simulate only one cirrus case. Nevertheless, the total number of simulated pixels including all wavelength channels and spatial resolutions is 313,000 for the 1-D simulations and 240,000 for the 3-D simulations. Note there are more 1-D computations because they are performed at

various scales from 50 m to 10 km while 3-D computations are only performed at 50 m.

## 3   Description of horizontal heterogeneity effects

Clouds have variabilities at many different scales. However, in retrieval algorithms, for simplicity and computational reasons, the independent column approximation (ICA; Stephens et al. (1991)) is commonly applied; cloud layers are assumed to be

vertically and horizontally homogeneous with an infinite horizontal extent (i.e. independent of each other). From the satellite retrieval point of view, the ICA is often named IPA for independent pixel approximation (Cahalan et al. (1994)). Obviously, in reality, the pixel is not homogeneous and the radiative transfer between cloudy columns occurs in 3-D. Comparisons of BT simulated with these two RT approaches (IPA and 3-D) allow us to highlight the cloud heterogeneity effects on BT.

We simulated BT with 3DMCPOL at scales ranging from 50 m to 10 km. For each scale, BT values are computed using the 1-D RT assumption at the averaged COT ($BT_{km}^{1D}$ with "x" the scale and "km" the distance unit) and compared with 3-D RT simulations at the finest field spatial resolution (50 m), averaged to the scale being considered. The latter are noted $\overline{BT_{50m}^{3D}}^{xkm}$. The choice of the 50 m spatial resolution corresponds to the highest spatial resolution that 3DMCPOL can achieve with a reasonable computational time for a 10 km domain.

In Fig. 5, we plot 1-D and 3-D BTs as a function of the optical thickness at the spatial resolution of 50 m (a), 250 m (b), 1 km (c) and 5 km (d) for the four MODIS TIR channels. For a better readability, 1-D cases are a color tone lighter than their corresponding 3-D case. First, we see that 3-D and 1-D BTs, decrease as optical thickness increases, because the warmer surface contributes less to the signal as the cloud becomes more opaque. Also, the relation between the BT and the

optical thickness is non-linear and depends on the optical thickness. This is particularly clear for the highest spatial resolution (50 m) where no aggregation of the 3-D BTs has been performed. Two effects explain the differences between 3-D and 1D BTs:

**Plane-parallel and homogeneous bias (PPHB):** The relation between BT and the optical thickness is nonlinear, leading to the Jensen inequality, and is usually referred to as the plane-parallel homogeneous approximation bias (PPHB, Cahalan et al.

(1994)). When BTs are aggregated from a high spatial resolution to a coarse spatial resolution, the average BT is different from the BT of the average optical thickness. In the thermal infrared, the averaged BT is larger than the BT directly computed from the average optical thickness. The PPHB is observed at all spatial resolutions (50 m, 250 m, 1 km and 5 km in Fig. 5 (a), (b), (c) and (d), respectively) and for decreasing resolution, the average BT3D is larger than the corresponding BT1D as predicted



by the Jensen inequality for the curvature of the relation.

For cirrus clouds observed in the thermal infrared from nadir, Fauchez et al. (2012, 2014) has shown that at a 1 km spatial resolution, the PPHB is the main heterogeneity effect and is essentially correlated (around 98%) with three parameters:

- The standard deviation $\sigma_\tau$ of the optical thickness inside the observation pixel.

- The brightness temperature contrast ($\Delta BT(CS - OP)$) between the clear sky (CS) and an opaque cloudy pixel (OP).

- The effective size of ice crystals (in the range $D_{eff} = 5 - 30 \ \mu m$ where the absorption varies significantly).

Because cirrus clouds can be very heterogeneous (Sassen and Cho (1992); Carlin et al. (2002); Lynch et al. (2002)) and their cloud top altitude very high (5 km to 20 km), the impact of the cloud horizontal heterogeneity on TOA BT can reach more

than 15 K for heterogeneous cirrus cloud pixels of $1 \times 1 \ km$ at about 10 km altitude (Fauchez et al. (2014)). It can probably be larger for tropical/equatorial cirrus for which the altitude can be higher than 15 km and the surface temperature larger than 310 K, as in this situation the contrast between the clear sky BT and opaque cloud pixel BT is large leading to a likewise large PPHB. Obviously, such a BT bias can severely impact a cloud optical property retrieval.

**Horizontal radiative transport (HRT):** In addition to the PPHB, the IPA error (IPAE) can also impact TOA BT through HRT. This effect is small in the TIR at a scale of 1 km but not necessary at a 50 m spatial resolution which is smaller than the FLIP mean path (distance traveled before absorption or cloud escape). Indeed, as seen in the different subplots of Fig. 5, 1-D calculations show a one-to-one relationship between BT and optical thickness, but the 3-D relation is highly dispersed because of HRT effects between cloudy columns. In addition, points are less scattered at the coarsest spatial resolutions (1 and 5 km)

which means that the HRT effect is reduced; the number of points are of course also reduced by the aggregation to coarser resolution.

In Fig. 6, we can see 3-D and 1-D BT, computed at 50 m resolution along a line parallel to the X-dimension in Fig. 3 (a) at a Y coordinate of 5 km, for channels centered at $8.52 \ \mu m$ and $13.36 \ \mu m$; also shown is the optical thickness at $12.03 \ \mu m$.

For both channels,1-D BT is larger than 3-D BT when the optical thickness is small and conversely when the optical thickness is large. This means that, on average, extreme values of 3-D BT are reduced by HRT smoothing. This effect is stronger at $8.52 \ \mu m$ where the cloud scattering is significantly larger and cloud absorption smaller. As a result the BT differences between 3-D and 1D are larger at $8.52 \ \mu m$ than at $13.36 \ \mu m$, particularly for large optical thicknesses, a tendency that will impact cloud optical property retrievals that use a combination of these channels.

Fig.7 shows the brightness temperature differences $\Delta BT = BT_{50m}^{3D} - BT_{50m}^{1D}$ and their distribution for each $50 \times 50$ m pixel of the $10 \times 10$ km field versus the number of pixels (top panels) and optical thickness (lower panels) for MODIS channels centered at $8.52 \ \mu m$ (a), $11.01 \ \mu m$ (b), $12.03 \ \mu m$ (c) and $13.36 \ \mu m$ (d). Positive values are shown in red, negative ones in blue.





Because BT values from 3-D and 1-D RT are computed at the same spatial resolution (50 m), there is no horizontal aggregation effect (no PPHB), only the HRT effect occurs. We can see that the largest values of $\Delta BT$ are at $8.52\ \mu m$ because of the larger single scattering albedo leading to more scattering. For this channel, $\Delta BT$ ranges from -9 K to +19 K (top panel in (a)) and is highly asymmetric regarding $\tau_{50m}^{12}$ (bottom panel in (a)). Indeed, *(i)* largest $\tau_{50m}^{12}$ preferentially lead to 3-D BT > 1-D BT

because, as seen in Fig.6 scattered FLIPs coming from small optical thicknesses (associated to largest BTs) drastically increase the BT of larger optical thicknesses through HRT. This effect is particularly noticeable for $\tau_{50m}^{12} > 6$ where only positives $\Delta BT$ exist. However, for very largest values, absorption is so strong that the $\Delta BT$ increase is mitigated. *(ii)* For the smallest $\tau_{50m}^{12}$ (below 3), negative $\Delta BT$ values dominate because fewer FLIPs come from thick and cold areas, decreasing the BT of these pixels (see Fig. 6). The minimum $\Delta BT$ is around $\tau_{50m}^{12} = 2$. Below this value, the efficiency of the HRT effect is reduced by

the decrease in cloud extinction. BTs are dominated by the surface emission, reducing the BT contrast between smaller and larger $\tau_{50m}^{12}$, and the chance of scattering is weak, leading to a small HRT effect.

The $\Delta BTs$ are smaller in channels at $11.01\ \mu m$, $12.03\ \mu m$ and $13.63\ \mu m$ and they are more symmetric with respect to 0 K. This greater symmetry is due to the smaller scattering and the larger absorption in the cloud (see optical properties in Table 2). This is particularly clear at $11.01\ \mu m$ where cloud extinction is significantly smaller than in the other channels, reducing

the probability of scattering, and thus the amplitude of the HRT effect, even if FLIPs can propagate farther in the cloud. For the three channels, below $\tau_{50m}^{12} = 3$, the HRT effect from large to small $\tau_{50m}^{12}$ pixels tends to dominate, leading in average to $(\overline{\Delta BT} < 0)$ but for larger optical thicknesses, it is the HRT effect from small to large $\tau_{50m}^{12}$ which dominates leading in average to $(\overline{\Delta BT} > 0)$. Contrary to the channel at $8.52\ \mu m$, FLIPs coming from small $\tau_{50m}^{12}$ propagate less to very large $\tau_{50m}^{12}$ because of the stronger absorption. In addition, the emission temperature between large optical thicknesses is quitte similar ($\sim 215$ K).

Therefore, very large $\tau_{50m}^{12}$ are hardly impacted by the FLIP transport (only $\pm 2$ K due to neighboring pixels with a similar $\tau_{50m}^{12}$). As a result, the maximum of $\Delta BT$ is around $\tau_{50m}^{12} = 5$. Note that for all the channels, the field-averaged error in $\overline{\Delta BT}$ due to HRT is almost nil. For the channel centered at $13.36\ \mu m$, $\overline{\Delta BT}$ is slightly positive (0.15 K) while for the others channels it is slightly negative ($\sim$ - 0.06 K). The reason why $\overline{\Delta BT}$ is positive at $13.36\ \mu m$ and negative for the others channels is due to the larger absorption optical thickness at $13.36\ \mu m$ causing a HRT effect dominated by the effect described above *(ii)*.

Obviously, both effects, the PPHB and HRT strongly depend on the observation scale discussed in the next section.

## 4   Horizontal heterogeneity effects as a function of the nadir observed scale

As discussed in Section 3, heterogeneity effects on the radiative fields observed from nadir at TOA depend, on the one hand, on the sub pixel optical thickness inhomogeneity (PPHB) and on the other hand, on the IPAE (HRT effect). The optimal resolution

for cloud retrievals is therefore a compromise between reducing the PPHB by improving the spatial resolution without causing larger increases in HRT effect. The objective is thus to find the smallest spatial resolution that strikes a balances between the PPHB and the absolute error due to the HRT. This spatial resolution depends of course on the wavelength (dependence on the FLIP mean path), cloud type (different optical properties, optical and geometrical thicknesses and altitude) and the geometry





of observation.

The total difference, computed as the total arithmetic mean difference (AMD) between aggregated 3-D and non-aggregated 1-D TOA BT viewed from nadir as a function of the spatial resolution, is given by:

$$AMD(\overline{\Delta BT_{xkm}^{3D-1D}}) = [\sum_{i=1}^{N}(\overline{BT_{50m}^{3D}}^{xkm} - BT_{xkm}^{1D})]/N, \qquad (2)$$

with $N$ the number of pixels at the spatial resolution $xkm$. 3-D BT at all scales are estimated by aggregating the 50 m BT to the $xkm$ scale, while 1-D BTs are directly computed at the $xkm$ scale after aggregating the 50 m optical thickness. Because averaging BTs from a fine to a coarser spatial resolution will give a different result than BTs of the averaged optical thickness, we thus compared here how the non linearity between brightness temperature and optical thickness, as well as 3-D radiative

effects, impact TOA BTs at a given spatial resolution.

In order to separate the contribution of the PPHB and HRT to the total AMD, we also aggregate the 50 m 1-D radiances to each xkm scale. The PPHB is then the arithmetic mean difference between aggregated 1-D and non-aggregated 1-D TOA BT viewed from nadir as a function of the spatial resolution, and is given by:

$$PPHB(\overline{\Delta BT_{xkm}^{1D-1D}}) = [\sum_{i=1}^{N}(\overline{BT_{50m}^{1D}}^{xkm} - BT_{xkm}^{1D})]/N, \qquad (3)$$

Note that, because the PPHB is always positive or nil, $\overline{\Delta BT_{xkm}^{1D-1D}}$ is also either positive or nil. It is straightforward to be positive for the whole field, but locally, at the scale of a pixel, it can be either positive or negative, contributing to increase or reduce the AMD.

To highlight the absolute effect of the HRT, which can be considered as the mean deviation of the BT due to HRT, we also calculate the total mean absolute difference (MAD) between aggregated 3-D and non aggregated 1-D TOA BT viewed from

nadir as a function of the spatial resolution using the following equation:

$$MAD(\overline{\Delta BT_{xkm}^{3D-1D}}) = [\sum_{i=1}^{N}(|\overline{BT_{50m}^{3D}}^{xkm} - BT_{xkm}^{1D}|)]/N, \qquad (4)$$

This is almost the same as Equation 2 but for the sum of the absolute value of the difference. The mean deviation due to HRT at each spatial resolution is then obtained by subtracting the PPHB from the total absolute mean difference MAD (|HRT|=MAD - PPHB). The MAD allows us to represent, at each spatial resolution, the mean deviation of the BT due to the cumulative effects

of PPHB and |HRT|, and it is this parameter that we seek to minimize in order to estimate the optimal pixel size for IR cirrus retrievals..





Figure 8 shows in (a) the AMD and MAD and (b) PPHB and |HRT| of $\overline{\Delta BT}$ estimated at TOA from nadir for the whole cirrus field as a function of the spatial resolution for the MODIS TIR channels centered at $8.52\ \mu m$, $11.01\ \mu m$ $12.03\ \mu m$ and $13.36\ \mu m$. In Fig. 8, we note that, for all channels, AMD is always smaller than MAD because the PPHB can be partially offset by the HRT when it is negative.

As the behavior is different for different channels, we discuss first the more absorbing channels (centered at $11.01\ \mu m$, $12.03\ \mu m$ and $13.36\ \mu m$) and then the more scattering channel (centered at $8.52\ \mu m$).

**Channels centered at $11.01\ \mu$m, $12.03\ \mu$m and $13.36\ \mu$m:**

In Fig. 8, it is evident that $\overline{\Delta BT}$ at $11.01\ \mu m$, $12.03\ \mu m$ and $13.36\ \mu m$ plotted as a function of the spatial resolution have approximately the same behavior because the optical properties of the cirrus at these wavelength are quite similar. As previously discussed, the largest heterogeneity bias for these channels is due to the PPHB (increasing with decreasing spatial resolution) leading to a maximum $\overline{\Delta BT}$ for the coarsest spatial resolution. In our case, at the spatial resolution of 10 km, the whole cirrus field is considered horizontally homogeneous, leading to the largest PPHB and AMD or MAD total biases. The differences

between the 3 channels are due to the differences in cloud absorption. Considering the optical properties in Table 2, $\overline{\Delta BT}$ increases with the absorption coefficient $\sigma_a$. Indeed, the PPHB increases with cloud absorption (Fauchez et al. (2012, 2014)) in the range where BT is a non-linear function of optical thickness $\tau$ ($0<\tau<10$ approximately). The larger AMD or MAD total biases are reached for the channel centered at $13.36\ \mu m$, with MAD at 10 km of about $\overline{\Delta BT} = 6.5\ K$. In fact, as the spatial resolution is improved from 10 to 2.5 km, the $\overline{\Delta BT}$ are quite stable as the field heterogeneity between these spatial resolutions

is similar (i.e. the number of fallstreak in a 2.5 km box is similar to the that of the whole field) . Note that, at these scales, the AMD or MAD total biases and PPHB are close and the HRT error approaches 0 because the FLIP mean path (FLIP average distance before absorption or before leaving the cloud) is much shorter than these scales.

$\overline{\Delta BT}$ AMD and MAD drastically change below 2.5 km where the PPHB rapidelly decreases with the improving spatial resolution. At 1 km, we can see that the |HRT| effect (Fig. 8 (b)) slightly increases, through this is more clearly visible at 500 m.

Between 250 m and 100 m, the HRT curves cross the PPHB curves and the HRT effect becomes the dominant effect. At 50 m, the PPHB is nil because this is the same spatial resolution as that of the model. However, the |HRT| effect is the largest at 50 m, because FLIPs can easily propagate through many small 50 m pixels. It is important to note that the competition between the two effects leads to a minimum overall MAD around 100 m for these 3 channels.

**Channel centered at $8.52\ \mu$m:**

The heterogeneity and horizontal transport effects on BT as a function of the spatial resolution have a very different behavior at $8.52\ \mu m$ due to a stronger cloud scattering. Indeed, in this channel, the single scattering albedo is about 0.3 larger than the value for the three others channels (see Table 2). A stronger cloud scattering has two consequences: *(i)* A smaller PPHB due to a decrease in cloud absorption and emission for an equivalent extinction. *(ii)* A larger IPAE due to an increase of |HRT|. Indeed,

we can see that, at 10 km, $\overline{\Delta BT}$ is equal to 2.1 K instead of the 4.2, 5.8 and 6.5 K for the channels centered at $11.01\ \mu m$,





12.03 $\mu m$ and 13.36 $\mu m$, respectively, implying that the PPHB is smaller. We can also see in Fig. 8 (b) that, similar to the three other channels, $\overline{\Delta BT}$ AMD or MAD are almost constant from 10 km to 2.5 km and the HRT effect is on average nil at these scales. But below 2.5 km, moving to higher spatial resolutions reduces the PPHB but increases |HRT| almost proportionally leading to a relatively stable MAD($\overline{\Delta BT}$). Netherless, we can see that the MAD minimum is located at about 250 m spatial

resolution which is a bit larger than for the three others channels because the stronger scattering effects and the weaker cloud absorption allow more FLIPs to propagate farther at 8.52 $\mu m$. Note that the $\overline{\Delta BT}$ values for the four channels are closer to each other for high than for coarse spatial resolutions. When the effects on BTs are roughly the same for all channels, the MAD($\overline{\Delta BT}$ impact on retrieved products may be mitigated (not show here).

To summarize, for this cirrus field, the best resolution for mitigating the cumulative effect of the homogeneous plane parallel bias and horizontal transport effect is about 100 m for the three channels with stronger cloud absorption, and is about 250 m for the channel centered at 8.52 $\mu m$.

By a quick and simple sensitivity study, we can simulate the inhomogeneity impact for various average cloud properties by

sampling the whole cloud scene according to three important parameters:

- The brightness temperature contrast between clear sky and opaque cloudy pixels.

- The average cloud optical thickness

- The average cloud heterogeneity

The increase of the brightness temperature contrast between clear sky and opaque cloudy pixels ($\Delta BT(CS-OP)$) increases

obviously the PPHB (larger nonlinear BT vs $\tau$ averaging effect) as well as the IPAE (HRT has a larger impact if columns have more different opacities). For example, Fauchez et al. (2014) has shown in their Fig. 16 that when increasing the cirrus top altitude from about 8 km to about 11 km, the total effect (AMD) on BT is multiplied by 3 for the channel centered at about 8 $\mu m$ and about 2.5 for the channels around 10 and 12 $\mu m$. These factors will of course depend on the cloud opacity and surface and atmospheric temperature. However the resolution of the minimum heterogeneity effect should be not affected by

a change of $\Delta BT(CS-OP)$ since only the FLIP energy will change but not its mean path. Considering the computational times involved we chose to rely on this hypothesis and not do other time consuming runs.

However, the impact of changing the two others parameters, the average cloud optical thickness and heterogeneity can be more easily tested by sampling the cloud pixels in different optical thickness or heterogeneity ranges. Indeed, for the 50 m ;

100 m ; 250 m ; 500 m ; 1 km ; 2.5 km ; 5 km and 10 km spatial resolutions correspond to 40, 000 ; 10,000 ; 1,600 ; 400 ; 100 ; 40 ; 16 and 1 pixels, respectively, which represent a large number of pixels with various optical thicknesses and heterogeneities.

For every spatial resolution, we decided pixels in four optical thickness $\tau$ categories:

- Small optical thicknesses: $\tau < 1.0$ [28, 735 pixels]





- Medium optical thicknesses: $1.0 \leqslant \tau < 3.0$ [17, 305 pixels]

- Large optical thicknesses: $3.0 \leqslant \tau < 6.0$ [5, 028 pixels]

- Very large optical thicknesses: $\tau \geqslant 6$ [1, 063 pixels]

Similarly, the optical thickness heterogeneity parameter $\rho_\tau = [StDev[\tau]/<\tau>]$ (Szczap et al. (2014)) is also sampled in
four ranges:

- Small optical thicknesses heterogeneity: $\rho_\tau < 0.3$ [ 8, 969 pixels]

- Medium optical thicknesses heterogeneity: $0.3 \leqslant \rho_\tau < 0.7$ [2, 724 pixels]

- Large optical thicknesses heterogeneity: $0.7 \leqslant \rho_\tau < 1.1$ [347 pixels]

- Very large optical thicknesses heterogeneity: $\rho_\tau \geqslant 1.1$ [89 pixels]

The results of this sensitivity study are presented in Fig. 9 for the optical thickness and Fig. 10 for the heterogeneity of the
optical thickness, and are summarized in Table 3. In this table, we can see the spatial resolutions where PPHB is larger than
the |HRT| effect, and vice versa, as well as the minimum of the total MAD effect. For clarity reason we chose to not show the
AMD and MAD values in on the figures, and to keep only the MAD values in the table. We can see that the change of the
optical thickness or the heterogeneity deeply affects the relative strength of the PPHB and |HRT|. As previously seen, the PPHB
dominates at large scales, while the |HRT| dominates at small scales, except for $\tau \geqslant 6$. Indeed, the PPHB increases with the
optical thickness while |HRT| decreases because of the increase of the cloud absorption. When the heterogeneity $\rho_\tau$ increases,
this allows the PPHB to increase and to be larger than the |HRT| even at finer spatial resolutions (roughly shifted from 250 m
to 100 m between $\rho_\tau<0.3$ and $\rho_\tau \geqslant 1.1$). Indeed, increasing $\rho_\tau$ leads, on average, to increase the optical thickness and thus the
cloud absorption which enhances the PPHB but mitigates the |HRT|. In addition, we can see that the spatial resolution where
the MAD is minimum is quite stable, for more clarity some values are highlighted in colors. Most of the time, this is at 100 m
spatial resolution (green), followed by 250 m (yellow) and 50 m (red). For the last one no clear conclusions can be drawn
because this is the smallest scale of the simulation. These conclusions are consistent with those of Fig. 8 for the whole cirrus
field. Note that in Fig. 9 and Fig. 10, the |HRT| can be negative in one specific optical thickness or heterogeneity range since
this is on the whole field, where the HRT is, on average, close to nil.

## 5   Heterogeneity effects as a function of the observation scale for off-nadir views

In the previous sections, results were shown for simulated observations from nadir. In this section, we discuss off-nadir obser-
vation geometries. In addition to the PPHB and HRT effects, another bias appears when looking off-nadir. Indeed, the oblique
line of sight can cross many different cloudy columns in 3-D radiative transfer mode, while in 1-D, the cloudy column under-
neath a given pixel is considered horizontally infinite and thus fully containing the line of sight. We name this last bias the





tilted homogeneous extinction assumption bias (THEAB). Note that the results of this section are strongly dependent on the cloud structure (with fallstreaks or not) and may be generalize to cirrus with similar patterns. Like the HRT effect, the THEAB is due to the IPAE and both effects are thus merged and represented in Fig. 11 by the IPAE. In Fig .11 we can see the AMD (bold lines with squares), the MAD (bold lines with triangles), the PPHB (dashed line with crosses) and the IPAE (doted line

with stars) of $(\overline{\Delta BT})$ in (a) for viewing zenith angles $\Theta_v = 0°$; $30°$; $60°$ at a viewing azimuth angle of $\Phi_v = 0°$, and (b) and (c) for viewing azimuth angles of $\Phi_v = 0°$; $45°$; $90°$; $180°$ at $\Theta_v = 30°$ and $60°$, respectively, as a function of the spatial resolution for the channel centered at $11.01\ \mu m$ only. Computations for other channels were too time expensive and a selection of a unique channel was preferred in order to highlight general behaviors related to off-nadir viewing geometries.

In Fig. 11 (a), $MAD(\overline{\Delta BT})$ at $\Theta_v = 30°$ and $60°$ for spatial resolutions between 1 and 10 km are larger than at nadir.

Indeed, the PPHB is enhanced due to the increasing of the curvature (non linearity) between BT and optical thickness with the view zenith angle as we can see in Fig. 12. Note that we can also see in this latter figure that the saturation in BT with respect to changes in optical thickness appears earlier at $\Theta_v = 60°$ ($\tau_{50m}^{1D} \sim 4$) than at $\Theta_v = 30°$ and $0°$ ($\tau_{50m}^{1D} \sim 8$) and $\Theta_v = 0°$ ($\tau_{50m}^{1D} \sim 4$).

In Fig. 11 (a), the mean absolute difference $MAD(\overline{\Delta BT})$ at $\Theta_v = 30°$ and $60°$ is very large below $\sim 1\ km$ due to the fact that

the line of sight crosses many different columns in 3-D (large THEAB, which contributes strongly to the IPAE represented by the dashed lines with stars). However, when looking at the arithmetic mean difference $AMD(\overline{\Delta BT})$, we see that at $\Theta_v = 60°$, it is negative for spatial resolutions below 500 m due to two effects *(i)* first, in 3-D, due to the very oblique view, the line of sight crosses many cloudy columns of various optical properties for which the extinction is summed and leading, in most cases, to large optical paths. Such large optical paths imply that the top of the cirrus mostly contributes to the TOA BT. In

contrast, some lines of sight cross through small optical thicknesses, letting FLIPs emitted from the surface (much warmer than the cloud top) contribute to the TOA BT. This leads to 3-D BT being smaller than 1-D BT in average, and thus to negative $AMD(\overline{\Delta BT})$ values. We can also see this in Fig. 13 (f) where the blue color (cold emission temperature at the cloud top) is more present than in 1-D (Fig. 13 (e)). *(ii)* in 3-D, the line of sight crosses so many different columns that, the difference between nearby lines of sight is reduced and the heterogeneity of the BT field is smaller in 3-D ($StDev[BT] \sim 20.2\ K$) than

in 1-D ($StDev[BT] \sim 22.3\ K$) as we can see in Fig. 13 (f) by comparison to Fig. 13 (e), respectively. Then, when 50 m BT values are aggregated following Equation 2, 1-D BT are increased more by the PPHB than 3-D BT are, which contributes to the overall tendency of 1-D BT > 3-D BT and to the negative value of $AMD(\overline{\Delta BT})$.

While both effects are particularly strong below about 500 m where the pixel size is small, they occur at every spatial resolution, explaining why PPHB is always larger than the AMD due to the IPAE. Note that, contrary to the PPHB, the THEAB

contribution to the IPAE does not increase monotonously with $\Theta_v$ because it is related to the heterogeneity of the extinction along the line of sight for 3-D computations at 50 m, which can be smaller at $\Theta_v = 60°$ than at $\Theta_v = 30°$. We can also see that, the IPAE is negative and the AMD is positive at 2.5, 5 and 10 km at $\Theta_v = 60°$. Knowing that the HRT effect does not impact BT because the pixel size is too large, the IPAE is essentially due to the THEAB. In contrast to the higher spatial resolutions, the number of cloudy columns crossed by the line of sight is small and the large aggregation homogenizes the field and thus

reduces the AMD to a level close to or equal to the MAD. As a result, the AMD is smaller than the PPHB which means that,





at coarse spatial resolutions, the PPHB clearly dominates and the AMD is reduced, and not amplified, by the IPAE. Since AMD=PPHB+IPAE, the IPAE is negative as PPHB is larger than AMD or MAD. For $\Theta_v = 60°$, the conclusions are similar to those for $\Theta_v = 30°$, but with larger differences due to the greater IPAE between 3-D and 1-D BTs.

Concerning the change of the viewing azimuth angle at $\Theta_v = 30°$ and $60°$, the difference of AMD, MA , PPHB and IPAE between the four angles is quite small except at $\Phi_v = 45°$. Indeed, at this azimuth angle, the lige of sight is parallel to the cirrus fallstreaks as we can see in Fig. 13 (g) and (h) for $\Theta_v = 60°$. Therefore, the variability along the oblique line of sight is weaker, reducing the smoothing effect of the 3-D field, which is closer to the 1-D field averaged heterogeneity ($StDev[BT] \sim 21.1\ K$ in 3-D and $StDev[BT] \sim 22.3\ K$) . In addition, the line of sight can pass only optically small paths and result in large BT just as in 1-D. As a result, $MAD(\overline{\Delta BT})$ at $\Phi_v = 45°$ is reduced at spatial resolutions where fall streaks are still observable ($\leq 2.5\ km$). Above this value, the spatial resolution is so low that the fall streaks are smoothed and the effect disappears.

Off-nadir, it is not obvious to determine the spatial resolution where the absolute value of $\overline{\Delta BT}$ due to the combined heterogeneity and 3-D effects reached a minimum because its depends on the viewing angle as well as on horizontal and vertical inhomogeneity. However, looking at Fig. 11 we can say that this location is at a coarser resolution than at nadir, as the THEAB drastically increases the $\overline{\Delta BT}$ especially at high spatial resolutions. The spatial resolution at nadir where the AMD of $\overline{\Delta BT}$ is the most mitigated for nadir view therefore sets a lower limit for off-nadir viewing geometries on both, the AMD of $\overline{\Delta BT}$ and the spatial resolution where the combined effects are minimum.

## 6 Conclusions

The accurate remote sensing of cirrus clouds is very important in order to improve the parametrization of clouds in climate models and to better understand their role in the Earth-atmosphere system. Cloud heterogeneities may have a significant impact on the accuracy of retrieved cloud optical properties. In this work, we model the impact of cirrus cloud heterogeneities on top-of-atmosphere (TOA) brightness temperatures as a function of the spatial resolution from 50 m to 10 km and at four MODIS thermal infrared channels centered at $8.52\ \mu m$, $11.01\ \mu m$, $12.03\ \mu m$ and $13.36\ \mu m$. A three-dimensional cirrus cloud structure is modeled with the 3DCLOUD cloud generator and radiative transfer simulations are performed with the 3DMCPOL code. In this study, we consider that the difference in nadir TOA thermal infrared brightness temperatures between 1-D RT inside a homogeneous cloudy pixel and 3-D RT inside a heterogeneous cloudy pixel with the same mean value come mainly from two effects: *(i)* the optical thickness horizontal inhomogeneity leading to the plane parallel approximation bias (PPHB) and the *(ii)* horizontal radiative transport (HRT) effect due to the independent pixel approximation error (IPAE). The cloud vertical heterogeneities of optical properties are here neglected, based on the findings of Fauchez et al. (2014, 2015). As previous studies already showed, the PPHB is the larger heterogeneity effect for nadir observations at the typical spatial resolution of polar orbiters such as AIRS, MODIS or IIR. The PPHB impacts mainly the low spatial resolutions while the IPAE impacts mainly the high spatial resolutions. Although, due to the IPAE, the amplitude of the error in BT can be large at high spatial resolutions, the difference between the errors for different channels is quite small in comparison to the difference at coarse resolution. A similar error between channels can then mitigate the impact on the optical property retrieval. For our simulated





cirrus case, we find that the approximate spatial resolution where the PPHB and HRT effects lead to a minimum total effect at nadir is between 100 m and 250 m. In order to extrapolate this result to different cirrus clouds, a sensitivity study has been conducted. The results show that changing the average cloud optical thickness affects the magnitude of the effects but does not significantly change the spatial resolution of the minimum. A space-born radiometer with a nadir spatial resolution between
100 m and 250 m will allow retrieval of cirrus optical properties in the thermal infrared with mitigated overall heterogeneity and radiative effects.

Concerning off-nadir views, when $\Theta_v > 0°$, the line of sight may crosses several different cloudy columns in 3-D RT but not in 1-D RT, leading to the tilted homogeneous extinction assumption bias, THEAB. This increases strongly the mean deviation between 3-D and 1-D BT, especially at fine spatial resolutions. However, in average, an increase in viewing zenith angle
decreases the 3-D BT values as well as their heterogeneity, reducing the total error due to PPHB and IPAE. The dependence of the total effect on the azimuth angle could also be important for particular viewing orientations with respect to the cloud. For instance, the cloud heterogeneity, and thus the total effect, is smaller when the line of sight is parallel to the fall streaks of the cloud, and is larger elsewhere. It thus seems that, for arithmetic field average values, the minimum total effect arrises at nadir. Also, the THEAB leads to a shift in the spatial resolution of the minimum total effect toward coarser spatial resolutions.
In Part 2 of this work we will study the impact of cirrus heterogeneities on visible and near infrared MODIS channels and will make comparisons with the result of this present Part 1. Additional perspectives will concern the impact of cirrus cloud heterogeneities on the optical property retrievals, as well as on the fluxes. Others clouds, such as cumulus or stratocumulus should also be considered, because results are expected to be strongly dependent on the cloud type.

## 7   Code availability

The simulated data used in this study were generated by the 3DCLOUD (Szczap et al. (2014)) and 3DMCPOL (Cornet et al. (2010), Fauchez et al. (2014)) closed-source codes. Please contact the authors for more informations.

*Acknowledgements.* The authors acknowledge the Universities Space Research Association (USRA) through the NASA Postdoctoral Program (NPP) for their financial support.

We thank Dr. Zhibo Zhang and the UMBC High Performance Computing Facility (HPCF) for the use of their computational resources
(MAYA). The facility is supported by the U.S. National Science Foundation through the MRI program (grant nos. CNS-0821258 and CNS-1228778) and the SCREMS program (grant no. DMS-0821311), with additional substantial support from the University of Maryland, Baltimore County (UMBC). See www.umbc.edu/hpcf for more information on HPCF and the projects using its resources.

We also thanks the NASA Center for Climate Simulation (NCCS) for the use of their computational resources (Discover).





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





**Figures**

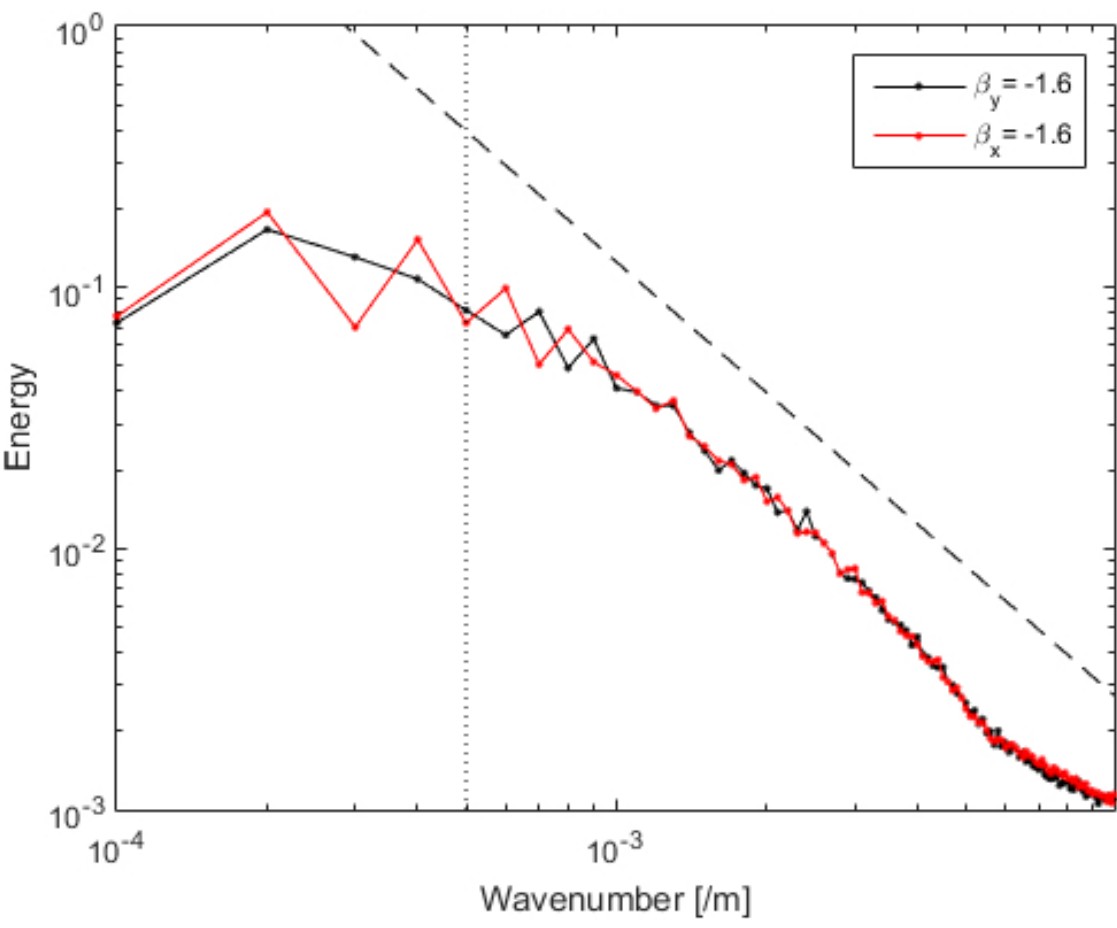

**Figure 1.** 1D averaged horizontal spectral slope of the "cirrus 1" case of study following the x axis (in red) and the y axis (in black). The spectral slope of a -5/3 theoretical signal is drawn (dashed line). Spectral slope values, between parenthesis, are estimated between $5.10^{-3}\ m^{-1}$ wavenumber (vertical dotted line) and the Nyquist wavenumber.





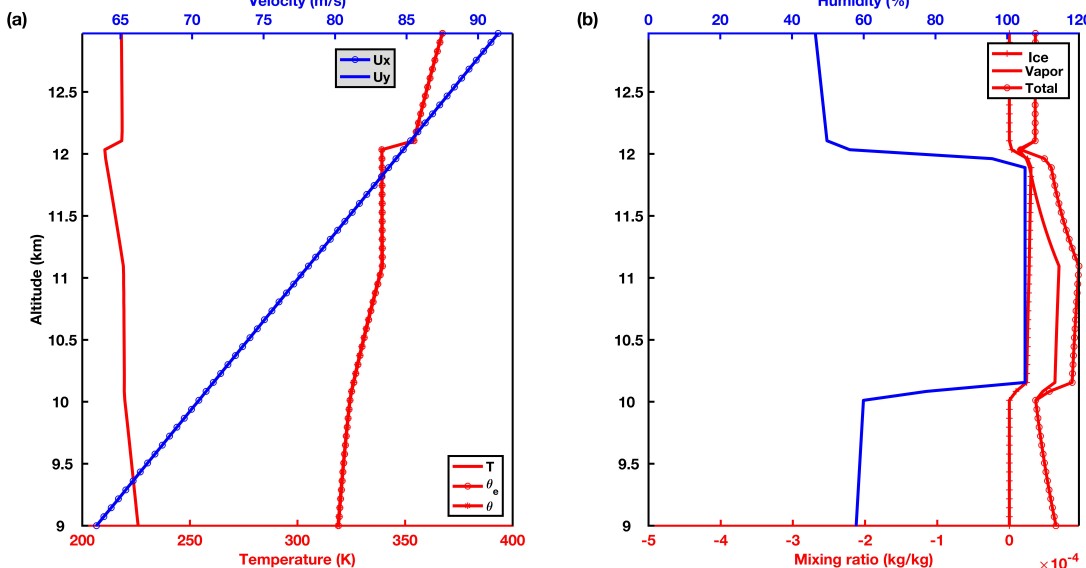

**Figure 2.** Meteorological profiles used to generate a realistic 3-D cirrus cloud field. (a) Wind velocity Ux and Uy (in blue) on the x and y axis respectively, temperature T, potential temperature $\theta$ and equivalent potential temperature $\theta_e$ (in red) as a function of the altitude, (b) relative humidity (in blue), ice, vapor and total mixing ratios (in red), as a function of the altitude. Note that Ux and Uy are over-imposed because the wind blow at $45°$ with respect to the x and y axis and that $\theta$ and $\theta_e$ are also over-imposed.



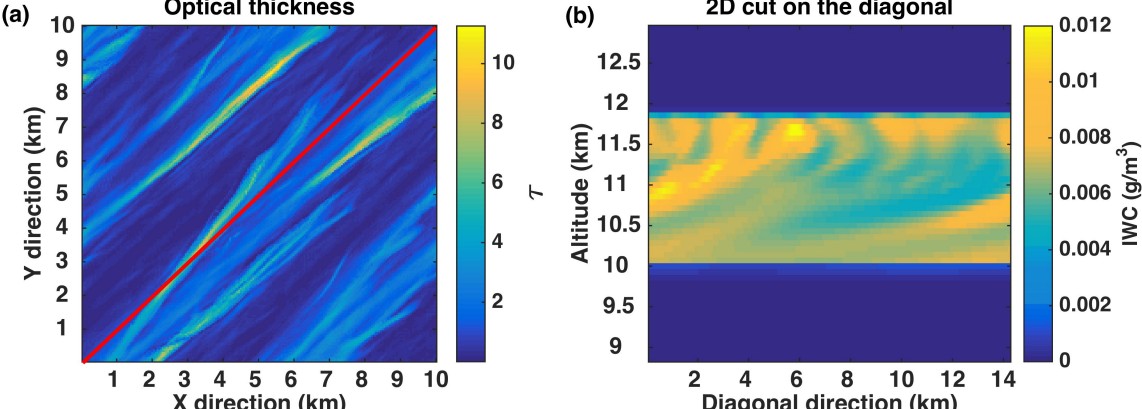

**Figure 3.** (a) $10 \times 10$ km optical thickness ($\tau$) field and (b) vertical cross section of ice water content (IWC) along the diagonal red line in (a). The mean optical thickness is 1.4 at 12.03 $\mu m$ and the heterogeneity parameter of the optical thickness is $\rho_\tau = 1.0$.





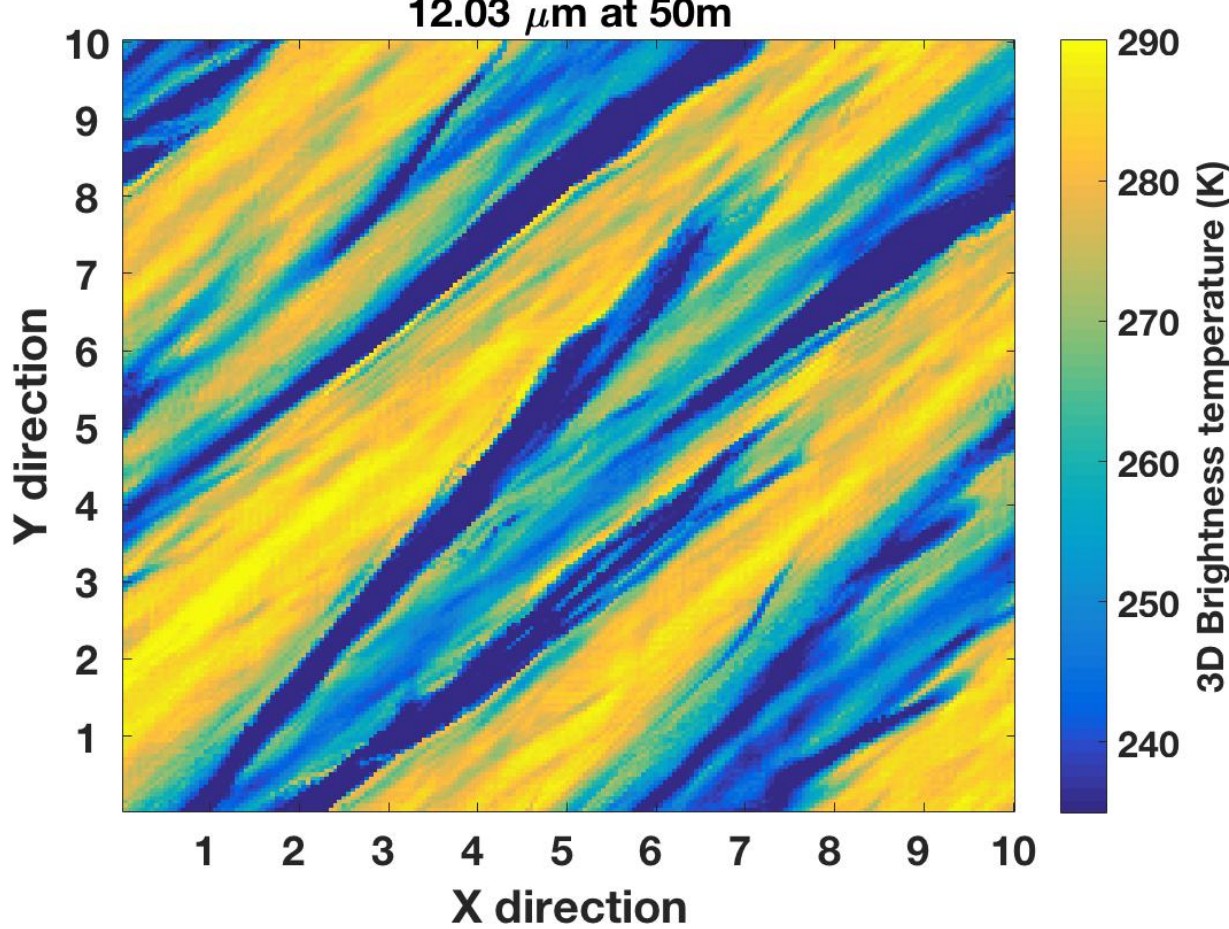

**Figure 4.** TOA brightness temperature field at 12.03 $\mu m$ of "cirrus 1".





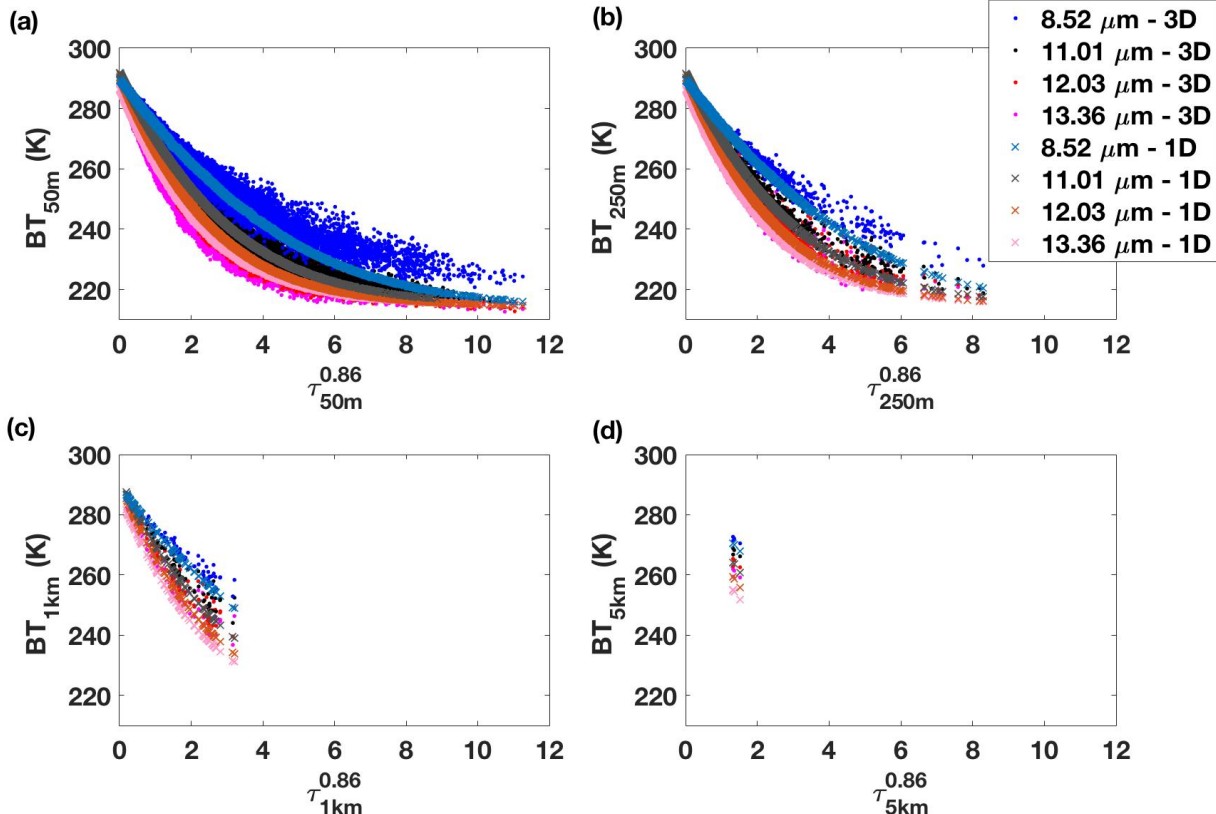

**Figure 5.** Brightness temperature (BT) as a function of the optical thickness at $0.86\ \mu m$ for MODIS channels centered at $8.52\ \mu m$, $11.01\ \mu m$, $12.03\ \mu m$ and $13.36\ \mu m$ at spatial resolutions of 50 m (a), 250 m (b), 1 km (c) and 5 km (d).




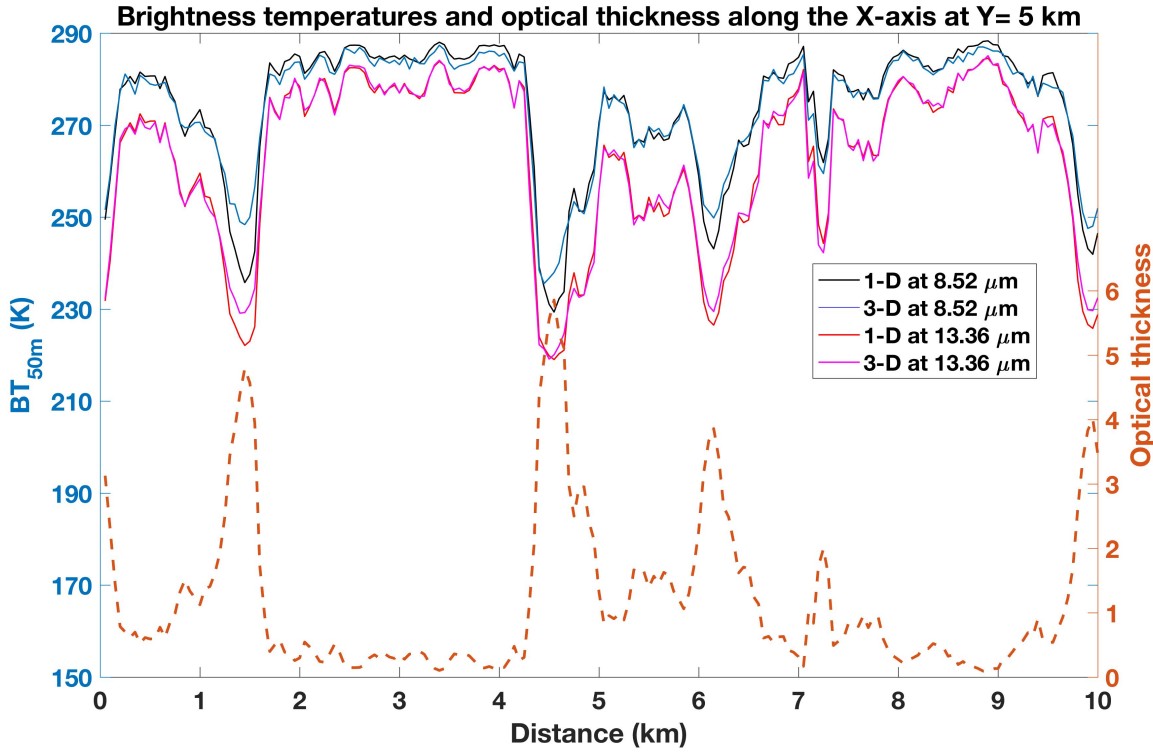

**Figure 6.** Optical thickness and brightness temperatures at 50 m spatial resolution computed in 3D ($BT_{50m}^{3D}$) at 8.52 $\mu m$, 11.01 $\mu m$, 12.03 $\mu m$ and 13.36 $\mu m$ along a line of constant Y axis coordinate (5 km) in the optical thickness field of Fig. 3 (a).





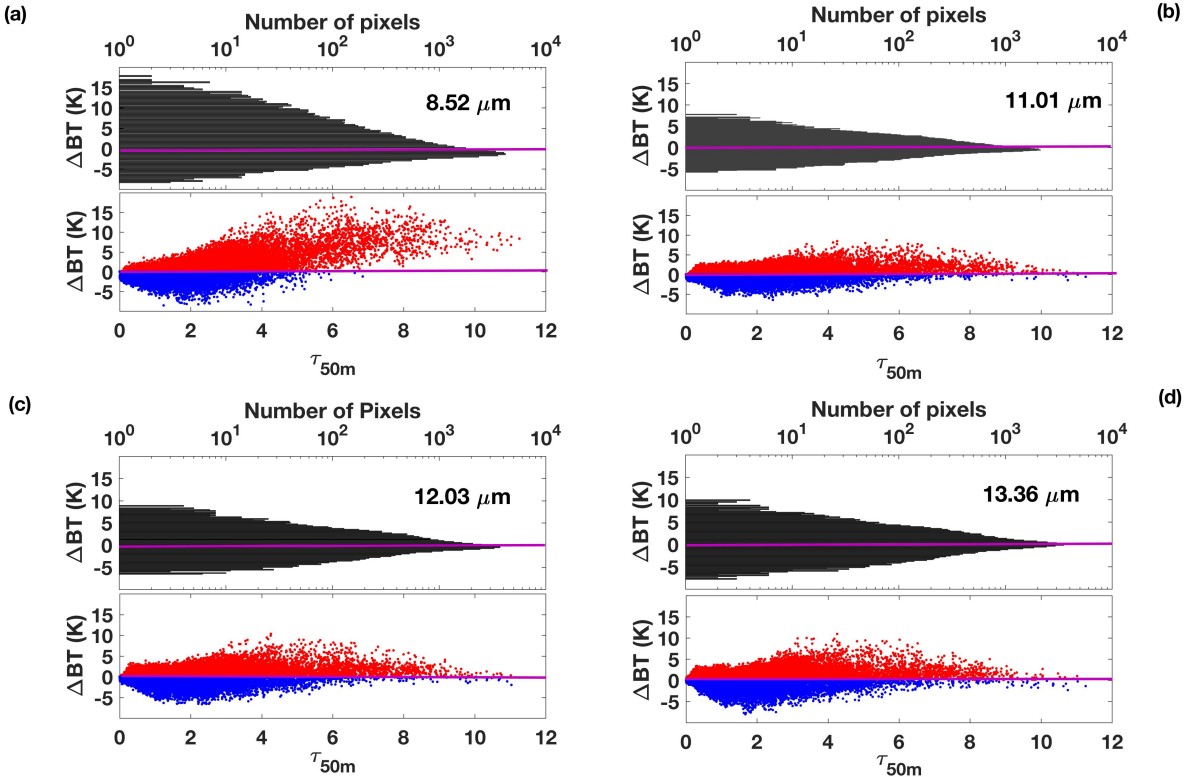

**Figure 7.** FLIP horizontal transport on TOA nadir brightness temperature differences ($\Delta BT = BT_{50m}^{3D} - BT_{50m}^{1D}$) as a function of the number of pixel and of the optical thickness at a resolution of 50 m. Positive and negative differences are in red and blue, respectively, at 8.52 $\mu m$ (a), 11.01 $\mu m$ (b), 12.03 $\mu m$ (c) and 13.36 $\mu m$ (d). For these four channels, the $\Delta BT$ percentage of positive values are, 33%, 40%, 41%, 53%, respectively.





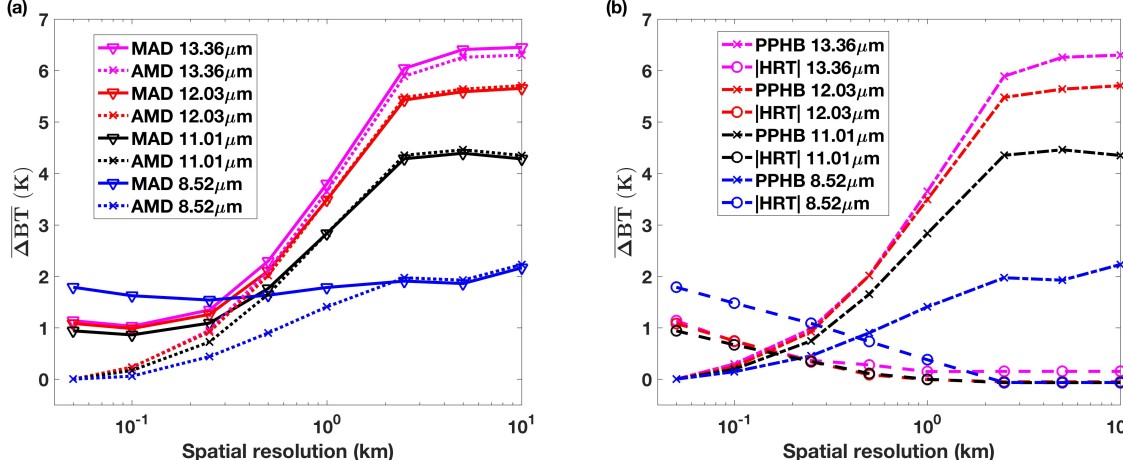

**Figure 8.** (a) Mean absolute difference (MAD) and arithmetic mean difference (AMD) between brightness temperatures computed in 3-D or 1-D following equation 2 and (b) plane parallel and homogeneous bias (PPHB) and mean deviation due to the horizontal radiative transport (|HRT|) on brightness temperatures as a function of the spatial resolution for channels at 8.52 $\mu m$, 11.01 $\mu m$, 12.03 $\mu m$ and 13.36 $\mu m$.




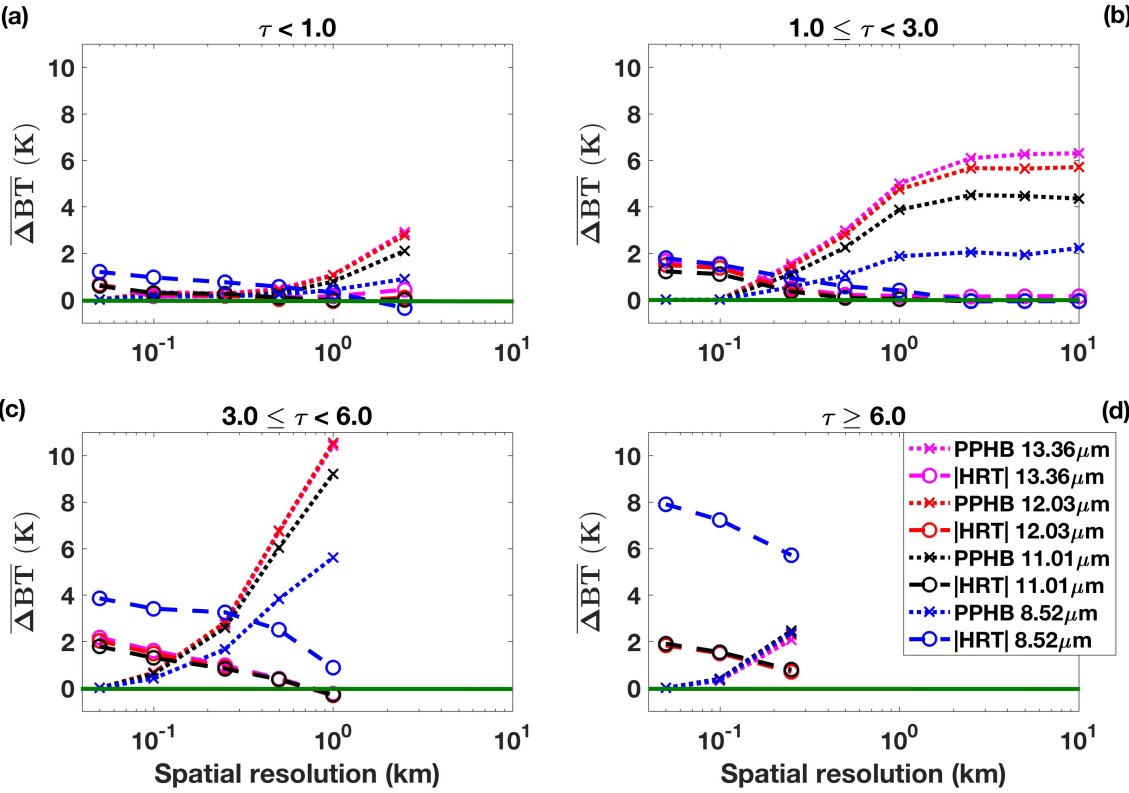

**Figure 9.** Scene average plane parallel and homogeneous bias (PPHB) and mean deviation due to the horizontal radiative transport (|HRT|) effect on brightness temperatures ($\overline{\Delta BT}$) for (a) small, (b) medium, (c) large and (d) very large pixel optical thicknesses as a function of spatial resolution in channels centered at $8.52\ \mu m$, $11.01\ \mu m$, $12.03\ \mu m$ and $13.36\ \mu m$. The small optical thickness range correspond to t 28, 735 pixels, the medium 17, 305 pixels, the large 5, 028 pixels and the very large 1, 063 pixels.





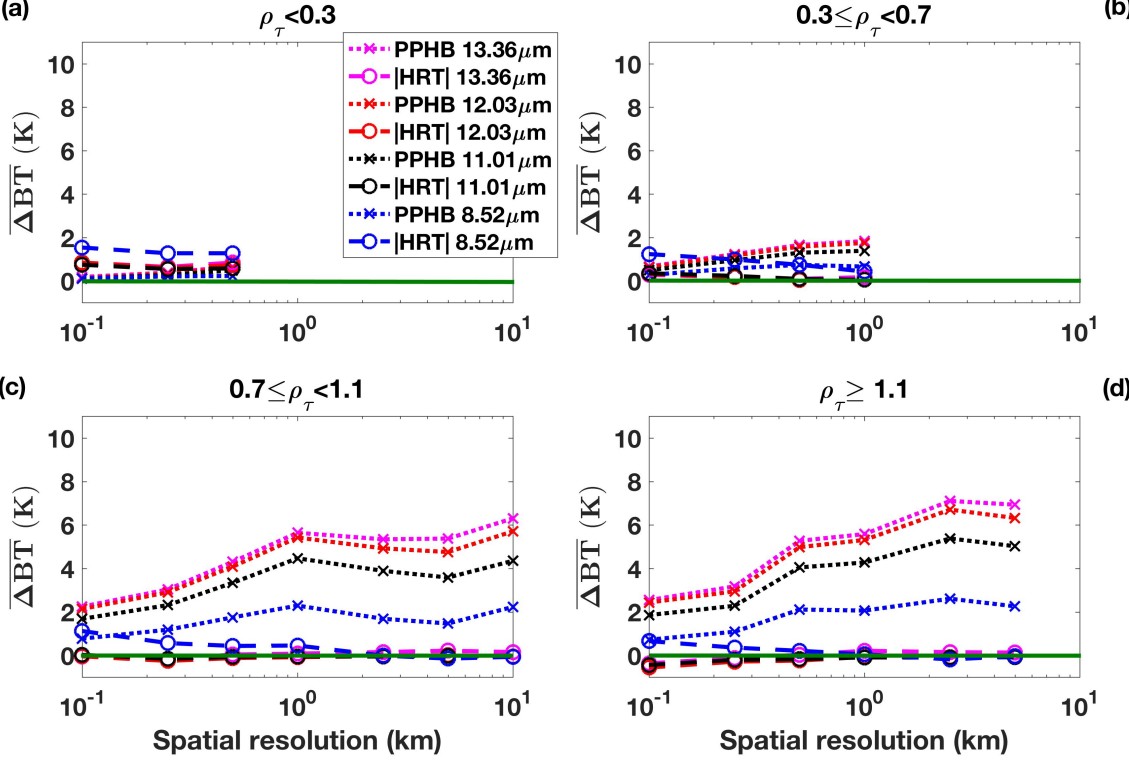

**Figure 10.** Scene average plane parallel and homogeneous bias (PPHB) and mean deviation due to the horizontal radiative transport (|HRT|) effect on brightness temperatures ($\overline{\Delta BT}$) for (a) small, (b) medium, (c) large and (d) very large pixel inhomogeneity ($\rho_\tau$) as a function of spatial resolution in channels centered at 8.52 $\mu m$, 11.01 $\mu m$, 12.03 $\mu m$ and 13.36 $\mu m$. The small optical thickness range correspond to 8, 969 pixels, the medium 2, 724 pixels, the large 347 pixels and the very large 89 pixels.





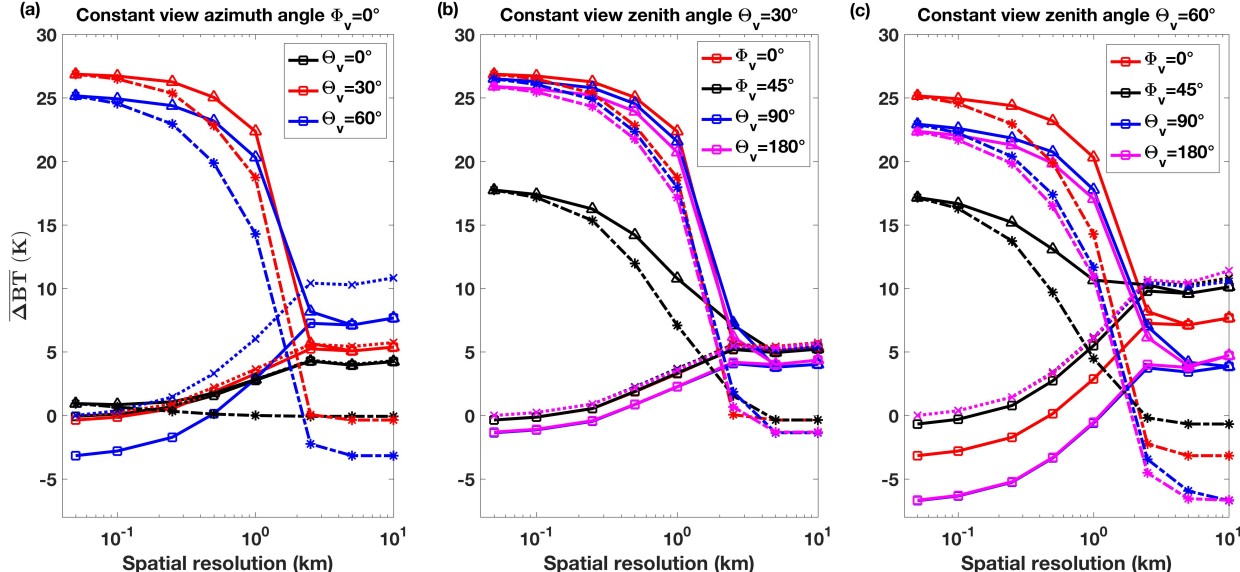

**Figure 11.** Mean absolute difference (MAD, lines with triangles), average mean difference (AMD, lines with squares) between 3-D and 1-D brightness temperatures estimated following equation 2, plane parallel and homogenous bias (PPHB, doted lines with crosses) and independent pixel approximation error (IPAE, dashed lines with stars) as a function of the spatial resolution for the channel centered at 11.01 $\mu m$ and as a function of (a) the viewing zenith angle $\Theta_v$ at an azimuth angle of $\Phi_v = 0°$ , (b) the viewing azimuth angle at $\Theta_v = 30°$ and (c) the viewing azimuth angle at $\Theta_v = 60°$.





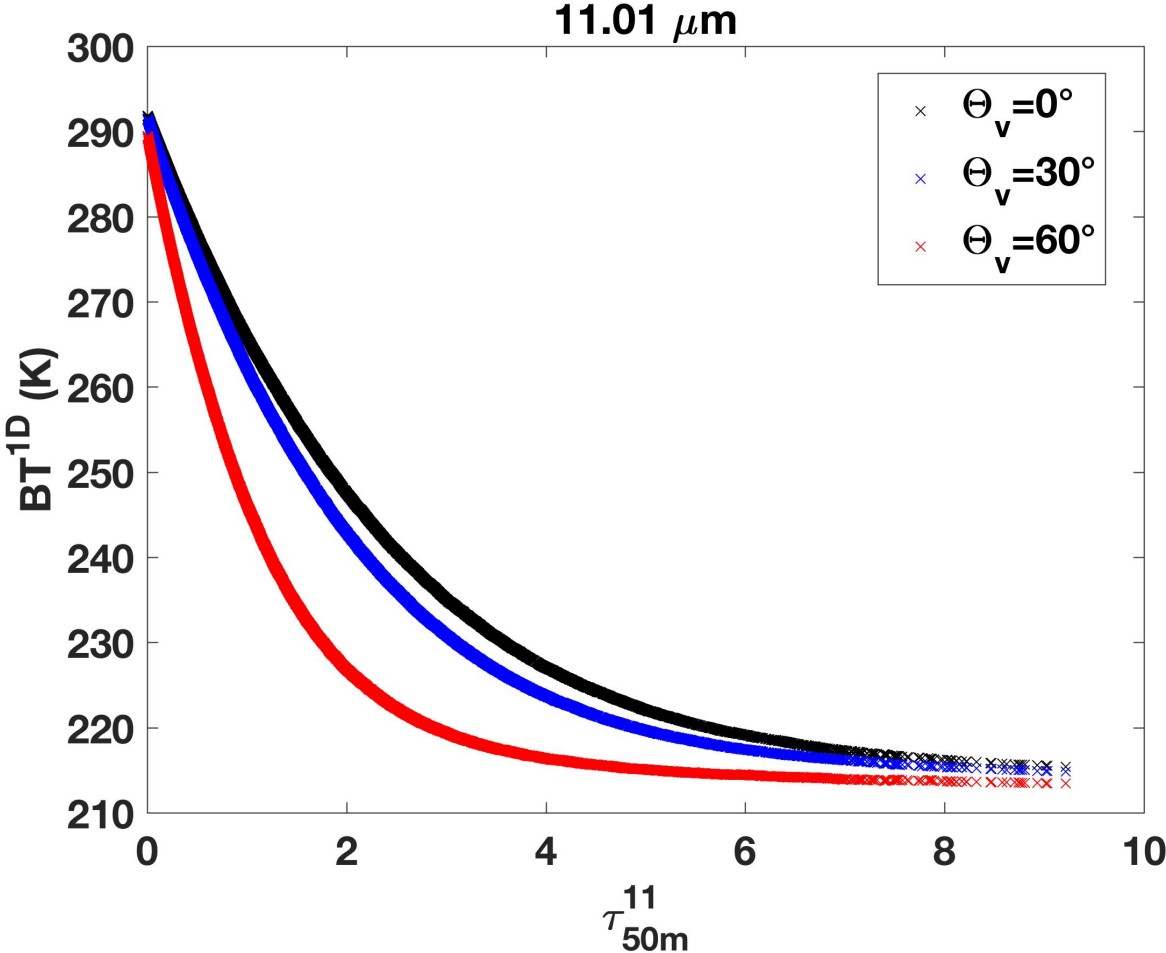

**Figure 12.** 1-D brightness temperatures as a function of the 50 m optical thickness $\tau_{50m}^{1D}$ for zenith view angles $\Theta_v = 0$, 30 and 60° at a constant view azimuth angle of $0°$.



**Figure 13.** 1-D ((a), (c), (e) and (g)) and 3-D ((b), (d), (f) and (h)) BT fields at $11.01\ \mu m$ and at 50 m spatial resolution view at a zenith angle of $\Theta_v = 0$, 30 and $60°$, respectively, for an azimuth viewing angle of $\Phi_v = 0°$ and $\Phi_v = 45°$ representing by the black arrows.



**Tables**



**Table 1.** Summary of key cirrus properties from the literature based on Dowling and Radke (1990), Sassen and Cho (1992), McFarquhar and Heymsfield (1997), Sassen et al. (2007, 2008), Szczap et al. (2014), Fu et al. (2000), Smith and DelGenio (2001), Buschmann et al. (2002), Carlin et al. (2002), Lynch et al. (2002), Hogan and Illingworth (2003), etc. For each property, the range of possible values, the mean value and the value of our simulation ("cirrus 1") are listed. Note that the optical thicknesses are given at $12\mu m$ and that the value in parenthesis corresponds to extreme optical thickness cases for cumulonimbus plumes.

| Properties | Range | Average | cirrus 1 |
|---|---|---|---|
| Geometrical thickness (km) | 0.1 - 8 | 2 | 2 |
| Cloud top altitude(km) | 4 - 20 | 9 | 12 |
| IWC $(g.m^{-3})$ | $10^{-4}$ - 1.2 | $2.5 \times 10^{-2}$ | $4.3 \times 10^{-3}$ |
| Crystal effective diameter $(\mu m)$ | 1 - 220 | 40 | 20 |
| Crystal shape | variable | variable | aggregate column |
| Optical thickness | 0.001 - 3(30) | 0.5 | 1.4 |
| Heterogeneity parameter of the optical thickness | 0.1 - 1.5 | 0.7 | 1.0 |





**Table 2.** Bulk scattering properties (extinction coefficient "$\sigma_e$", absorption coefficient "$\sigma_a$", single scattering albedo "$\varpi_0$" and asymmetry parameter "$g$") of the aggregate column ice crystal (Yang et al. (2013)) with an effective diameter of 20 $\mu m$, for the four channels use in this study.

| | $\sigma_e$ [$km^{-1}$] | $\sigma_a$ [$km^{-1}$] | $\varpi_0$ | g |
|---|---|---|---|---|
| MODIS channel 29 (8.52 $\mu m$) | 2.346646 | 0.594559 | 0.7466347 | 0.8643211 |
| MODIS channel 31 (11.01 $\mu m$) | 1.599258 | 0.922958 | 0.4228833 | 0.9313643 |
| MODIS channel 32 (12.03 $\mu m$) | 1.954191 | 1.028474 | 0.4737085 | 0.9126511 |
| MODIS channel 33 (13.36 $\mu m$) | 2.145600 | 1.062924 | 0.5046031 | 0.8995098 |



**Table 3.** Ranges of spatial resolutions from which PPHB is larger than mean deviation due to HRT (|HRT|), |HRT| larger than PPHB and where the total absolute difference is minimum (MAD min), for different range of optical thickness $\tau$ and heterogeneity parameter $\rho_\tau$ for four MODIS TIR channels. ($\overline{\rho_\tau} = 1.0$) and ($\overline{\tau} = 1.4$) are the average values on the whole cirrus corresponding to the different ranges of $\tau$ and $\rho_\tau$, respectively. See also Fig. 9 and Fig. 10.

| Range | | $\tau < 2.0$ | $2.0 \leq \tau < 3.0$ | $3.0 \leq \tau < 6.0$ | $\tau \geq 6.0$ | $\rho_\tau < 0.3$ | $0.3 \leq \rho_\tau < 0.7$ | $0.7 \leq \rho_\tau < 1.1$ | $\rho_\tau \geq 1.1$ |
|---|---|---|---|---|---|---|---|---|---|
| | | ($\overline{\rho_\tau} = 1.0$) | | | | ($\overline{\tau} = 1.4$) | | | |
| Effect | Channel | Spatial resolution range | | | | | | | |
| PPHB>|HRT| | 8.52 $\mu m$ | $\geq 2.5km$ | $\geq 250m$ | $\geq 100m$ | $\geq 100m$ | $\geq 250m$ | $\geq 1km$ | $\geq 250m$ | $\geq 250m$ |
| | 11.01 $\mu m$ | $\geq 500m$ | $\geq 100m$ | $\geq 50m$ | $\geq 50-100m$ | $\geq 250m$ | $\geq 100m$ | $\geq 100m$ | $\geq 100m$ |
| | 12.03 $\mu m$ | $\geq 250m$ | $\geq 100m$ | $\geq 50m$ | $\geq 50-100m$ | $\geq 250m$ | $\geq 100m$ | $\geq 100m$ | $\geq 100m$ |
| | 13.36 $\mu m$ | $\geq 250m$ | $\geq 100m$ | $\geq 50m$ | $\geq 50-100m$ | $\geq 250m$ | $\geq 100m$ | $\geq 100m$ | $\geq 100m$ |
| |HRT|>PPHB | 8.52 $\mu m$ | $\leq 2.5km$ | $\leq 250m$ | $\leq 100m$ | $\leq 100m$ | $\leq 250m$ | $\leq 1km$ | $\leq 250m$ | $\leq 250m$ |
| | 11.01 $\mu m$ | $\leq 500m$ | $\leq 100m$ | $\leq 50m$ | $\leq 50-100m$ | $\leq 250m$ | $\leq 100m$ | $\leq 100m$ | $\leq 100m$ |
| | 12.03 $\mu m$ | $\leq 250m$ | $\leq 100m$ | $\leq 50m$ | $\leq 50-100m$ | $\leq 250m$ | $\leq 100m$ | $\leq 100m$ | $\leq 100m$ |
| | 13.36 $\mu m$ | $\leq 250m$ | $\leq 100m$ | $\leq 50m$ | $\leq 50-100m$ | $\leq 250m$ | $\leq 100m$ | $\leq 100m$ | $\leq 100m$ |
| MAD min | 8.52 $\mu m$ | $\sim 2.5km$ | $\sim 250m$ | $\sim 100m$ | $\sim 100m$ | $\sim 250m$ | $\sim 500m$ | $\sim 250m$ | $\sim 250m$ |
| | 11.01 $\mu m$ | $\sim 500m$ | $\sim 100m$ | $\leq 50m$ | $\leq 50m$ | $\sim 250m$ | $\leq 100m$ | $\leq 100m$ | $\leq 100m$ |
| | 12.03 $\mu m$ | $\sim 250m$ | $\sim 100m$ | $\leq 50m$ | $\leq 50m$ | $\sim 250m$ | $\leq 100m$ | $\leq 100m$ | $\leq 100m$ |
| | 13.36 $\mu m$ | $\sim 250m$ | $\sim 100m$ | $\leq 50m$ | $\leq 50m$ | $\sim 100m$ | $\leq 100m$ | $\leq 100m$ | $\leq 100m$ |