# Peer review of "Scale dependence of cirrus horizontal heterogeneity effects on TOA measurements."

_Atmospheric Chemistry and Physics, 2017_

## Referee Comment (RC1) · Anonymous Referee #1 · 11 Mar 2017

**General comments**

The paper compares 3D and 1D Monte Carlo simulations of a cirrus cloud field at four different MODIS wavelength channels in the thermal spectral range. The aim of the study is to investigate the difference in brightness temperature between 1D and 3D radiative transfer in an inhomogeneous cloud field from a nadir satellite perspective and to find the optimal horizontal resolution where the error between the realistic 3D radiative transfer and the commonly used 1D approximations are at a minimum. Simulations of different horizontal resolutions (50m to 10km) have been performed and differences due to horizontal transport of radiation and the averaging/aggregation of high resolution pixels to coarser resolution, the plane parallel bias, have been addressed.

It was shown that the optimal horizontal resolution varies between 100 and 250m, depending on the wavelength channel. Even at this optimal resolution the difference in brightness temperatures between the 1D and 3D radiative transfer simulation can be up to 7K.

Additionally, sensitivity tests for varying optical properties have been performed.

The off nadir perspective was addressed by simulating one of the four MODIS channels of this study.

With this study, the authors extend former work in this field by showing the difference between 1D and 3D RT brightness temperatures at different horizontal resolutions. The paper is suitable for publication after minor revision.

General Comments:

The optical thickness used in the paper is not always defined. In one figure, the optical thickness at $0.86\mu$m is shown, while most of the manuscript refers to the $12.03\mu$m optical thickness. It is not always mentioned which optical thickness is used for the comparisons. The authors might clarify which optical thickness is used where in the study. I would recommend using a single one. What is the reason for choosing that specific wavelength optical thickness?

The authors refer often to the mean path of a photon/FLIP when effects of the horizontal resolution are concerned. It might help readers to have a certain number associated with the mean path at the four different wavelengths considered in the work. Maybe the value of the mean path at a certain optical thickness (e.g. 1 or 1.4 as this seems to be the mean optical thickness of the cirrus cloud in this study) could be added.

Some of the figures are hard to read. Especially the choice of red and pink in many of the line plots make it difficult to see the difference in the results. Please see the more specific comments below.

Many abbreviations are introduced in the introduction. Sometimes the authors use capital letters to show the origin of the abbreviation, but not throughout the text. I recommend doing this throughout the text.

How much different would results of a simulation of the 8.52$\mu$m channel in the off nadir perspective be? As this channel has a stronger scattering, one might expect stronger 3D effects? I understand that these simulations are expensive, but it might be worth adding this channel to the analysis, or discuss possible differences in the results.

The 'Conclusion' in its current form is a summary of the shown work. An outlook and some discussion about the implications of the results is wanting. Please see the more specific comment below.

**Specific comments**

1) Page 2, Line 1: Delete "due"

2) Page 2, Line 6: Change "of their optical properties" to "cirrus cloud optical properties"

3) Page 2, Line 15-18: This part is challenging to read and understand. I guess that the authors want to point out that the thermal infrared spectral range should (next to the retrieval of temperature/pressure and altitude) also be used for the retrieval of

optical properties such as COD and CED? This is part of the motivation for the study and should be pointed out more clearly.

4) Page 2, Line 19: Comma is missing: "AVHRR, "

5) Page 2, Line 20: delete brackets: ((Garnier et al., 2012, 2013))

6) Page 2, Line 21: example concerning the capital letters mentioned above: "Optimal Estimation Method" (OEM)

7) Page 2, Line 28: "etc.": The authors might add additional reasons or change the sentence to: "due to time constraints on 3-D forward radiative calculations and the lack of. . . "

8) Page 3, Line 1: Is longwave here the same as thermal IR?

9) Page 3, Line 1: Is the cooling rate in 1D too high or too low by 10%?

10) Page 3, Line 13: delete PPHB; it is already introduced at this point.

11) Page 5, Line 27: optical thickness at which wavelength?

12) Page 6, Line 10: Delete sentence "Note that TIR retrieval techniques are often limited to effective diameters between 5 and 50$\mu$m." either here or in line 5/6 above.

[Figure]

13) Page 6, Line 30: "cirrus 1" - There is only one cirrus case used in this study. I recommend deleting "cirrus 1" in the whole manuscript. Otherwise one would expect more than one scene.

14) Page 7, Line 18/19: The authors might mention the FLIP mean path as a second motivation for the 50m resolutions already at this point. I saw that it is mentioned later in the text, but it would already be worthy here.

15) Page 7, Line 21: Mention the wavelength of the optical thickness here. From Figure I take that it is at $0.86\mu$m. Why? If the optical thickness is taken in the visible, the 550nm is a common wavelength to use. For the rest of the paper the authors use the $12.03\mu$m optical thickness. I suggest using the $12.03\mu$m here as well. Additionally, why is the $12.03\mu$m wavelength chosen? It is one of the channels of course, but how strong does the optical thickness vary for the wavelength of the other channels?

16) Page 8, Line 26-28: Something about this paragraph is confusing and requires a better explanation. After reading it several times, I still cannot understand it in full. You point out that extreme values of the BT are smoothed out by the HRT effect. Therefore the difference between 1D and 3D BT should be smaller. As there is more scattering in channel $8.52\mu$m, one would expect smaller differences between 1D and 3D BT from the first conclusion. However, Figure 6 and your text show the opposite. This paragraph needs clarification. In addition, the choice of colors, the thickness of the lines and the scale of the y-axis makes this figure hard to read.

17) Page 9, Line 13: replace "smaller scattering" by "less scattering"

18) Page 9, Line 19: typo: quite instead of quitte

19) Page 11, Line 21: delete "(FLIP average distance before absorption or before leaving the cloud)" - this is explained a few times already

20) Page 11, Line 22: typo: rapidly instead of rapidelly

21) Page 12, Line 4: typo: Nevertheless instead of Netherless

22) Page 12, Line 32: optical thickness at which wavelength?

23) Page 13, Line 12: "we chose to not show" – replace by "we chose not to show"

24) Page 14, Line 7: I fully understand that Monte Carlo simulations are very expensive in terms of computational time. However, as scattering is stronger in the $8.52\mu$m channel and more horizontal transport of FLIPs between the column should occur, it might be worth adding this channel to the analysis? What result would be expected for the $8.52\mu$m channel?

25) Page 14, Lines 20-22: Reformulate this sentence "In contrast, some lines of sight cross through small optical thicknesses... "

26) Page 14, Line 28: Remove "about"

27) Page 15, Line 5: typo: lige

28) Page 15, Line 24-26: Reformulate sentence: "In this study, we consider..."

29) Conclusion: An outlook concerning the presented work would be beneficial for this section. The authors briefly state what will be shown in a Part 2 paper, however different wavelength channels are involved there.
As the choice of the cloud scene seems to have a larger impact on the off-nadir results, additional simulations (in future work) including different cirrus cloud fields might be one aspect. In addition, some discussion about the implications of the results for current cirrus cloud retrievals is wanting. How much would a satellite instrument with a resolution of 100-250m improve current retrievals? One might discuss that in the context of earlier studies (e.g. Fauchez et al., 2015) where the BT differences of 10K was related to ice crystal diameter and retrieved optical thickness. Is there a guess how much this improved resolution, with the following smaller differences in BT would improve the retrieval results? Currently, the conclusion section does not really show any conclusions. It only summarizes the presented work.

30) Figure 1, Caption: Delete "cirrus 1" and add "of the study"

31) Figure 2: Is the potential temperature and the equivalent potential temperature really the same?

32) Figure 3: Which optical thickness is shown in the figure?

33) Figure 4: Delete "cirrus 1"

34) Figure 5: Why do you use the optical thickness at $0.86\mu$m here and $12.03\mu$m in the
following? The colors and especially the markers are hard to separate in this figure. One really has to zoom into the pdf.

35) Figure 6: The difference between the lines is hard to see, especially the red and pink colors are hard to differentiate. Also, the scale of the y-axis makes it difficult to see the differences properly. The authors might also consider plotting thicker lines.

36) Figure 7: The first sentence of the caption is challenging to understand.

37) Figure 8 and following: Please use a different color for the pink lines. Maybe green or orange?

38) Figure 9: delete the "t" after "to" at the end of he third line.

39) Figure 10: The lines in the upper row are hard to separate. I can see that you want to keep the values of the y-axis constant, but you might think of reducing it to 8 instead of 10? Maybe this would already help?

40) Table 1: remove "cirrus 1"

**Technical corrections**

Please see the "Specific Comments" section.

---

## Referee Comment (RC2) · Anonymous Referee #2 · 19 Mar 2017

The authors present the impact of the horizontal heterogeneity of cirrus clouds on TOA brightness temperatures for 4 TIR MODIS channels. The study is based on a "realistic" cirrus case simulated using the 3DCLOUD model, MODIS Collection 6 ice crystal properties, and the 3DMCPOL radiative transfer code. This study discusses the impact of the plane parallel homogeneous bias (PPHB) and of the horizontal radiative transport (HRT) in various conditions of optical depth, optical depth inhomogeneity, and viewing angles. The paper also discusses the optimum horizontal resolution that minimizes the horizontal heterogeneity effects on TOA brightness temperature.

General comments:

The simulations and the results are solid. The simulated cirrus case is well adapted to illustrate the PPHB and the HRT. However, the impact of this choice on the conclusions

of the paper should be discussed. It would be important to know to what extent these results could be generalized. The main characteristics of the simulated cloud should be given in the abstract (lines 7-9).

The reasoning and the story are sometimes difficult to follow. Introductory and linking sentences would be sometimes helpful for the clarity of the manuscript.

My recommendation is to publish this manuscript after clarification on the several points listed above and hereafter.

1)Title:

The title could specify that this paper discusses cirrus heterogeneity effects on TOA brightness temperatures. "cirrus heterogeneity effects" is too vague, in my opinion.

2)Goal of the paper:

Page 3, lines 17 to 21: Please explain the choice of these 4 TIR channels. In which MODIS algorithm(s) are they used and what are the retrieved geophysical parameters? ". . .. the impact of horizontal heterogeneity. . ." Please specify impact on which quantity (TOA BT, optical depth, CED, other?).

3)Realistic cirrus case:

The rationale for the choice of the "realistic" cirrus case should be clearly presented. Table 1 should be presented and discussed in more detail. I agree that assuming a "constant" CED of 20 $\mu$m (page 6, lines 9-12) is "realistic", but it is not typical nor statistically representative. The fact that TIR techniques are often limited to CED between 5 and 50 $\mu$m (page 6, line 10) clearly does not mean that all CED are so small (as shown in Table 1). Please clarify the rationale.

The impact of this choice on the conclusions of the paper should be discussed. In particular, how does it impact the highlighted difference between the 8.52 $\mu$m channel and the three other channels?

[Figure]

Page1, line 7: "A unique but realistic cirrus case is simulated...": Why is the cirrus case "unique"? Do you mean that only one case is simulated?

4)Averaging and aggregation:

Please define "averaging" and "aggregation", and use consistent terms throughout the paper. Below are some examples (there are more in the text):

Page 7: line 17: "...averaged to the scale being considered...". Please detail the averaging process. Which parameter?

Page 7, line 26: "..aggregation.." Please explain what "aggregation" means.

Page 7, line 30 : "..the averaged BT.." Are you averaging BT? I am surprised because the observations are radiances (same comment page 10, line 8).

Page 10, line 7: ", while 1-D BTs are directly computed at the xkm scale after aggregating the 50 m optical thickness" My understanding is that 1-D BT are computed using an averaged optical depth. Is is what you mean?

5)Other comments (mostly for clarification):

Page 3, lines 24-25: '"we describe the heterogeneity and 3-D effects" For more clarity, it is suggested to specify PPHB and IPAE (or horizontal radiative transport).

Page 5, line 9: Figure 1, caption: what is 'Cirrus 1"?

Page 5, line 29: "For the cirrus used in this study..." Is it cirrus 1 listed in Table 1? Please clarify. Introduce Table 1 earlier. The references listed in Table 1 should be presented and discussed in the text.

Page 5, line 34: '....vertical variability of the geometrical and optical thickness.." Please clarify. I don't understand the notion of vertical variability of such quantities.

Page 6, line 3: for more clarity, title of Sect. 2.2 could be "ice crystal optical properties".

Page 6, line 4: "cirrus optical property parametrization": not entirely clear to me...what

about "bulk scattering properties? Is there really a parametrization?

Page 6, lines 5-6: "Note that TIR....between 5 and 50 $\mu$m". Why this sentence here?

Page 6, lines 7- 9: "...Holz et al. (2015) better consistency between ....the IR-split-window technique....and (VNIR/SWIR/MWIR) techniques, as well as with lidar retrievals........". This sentence is very confusing and I do not think that it is entirely correct. You are talking about the consistency between techniques and retrievals. Are you talking about retrieval of optical depth, or CED, or both? "Split-window technique" suggests CED. "Lidar retrievals" suggests "optical depth". Holz et al. (2015) discuss only optical depths, but not CED. Please clarify.

Page 6, line 32: "... as will be explained..." Specify in which section.

Page 7, line 21: Figure 5 According to the caption, this is now optical depth at 0.86 $\mu$m not introduced earlier. Please explain.

Page 7, line 33: "decreasing" resolution can be misunderstood. The notion of coarse or fine resolution would avoid any confusion.

Page 8, lines 8-13: The authors are discussing Fig. 5, and I am surprised to find these 6 lines with results from another paper. Why not discuss BT 3-D – BT 1D from Fig. 5?

HRT section: please re-organize the text for more clarity. - Lines 1-2 page 9 (HRT effect only when BT from 3-D and 1-D at the same resolution of 50 m) should be at the beginning of this sub-section, because important for a good understanding of the discussion. - Figure 6: it is suggested to add arrows to point to the areas of specific interest discussed in the text. A second panel showing BT differences between 3-D and 1-D could be helpful. - page 8, line 29: can you give an example of cloud optical property retrievals that use a combination of the 8.52 $\mu$m and 13.36 $\mu$m channels? - Figure 6, caption: I don't see the BTs computed at 11.01 and 12.03 $\mu$m. Lines 5-6, page 9 ("as seen in Fig. 6...") could be useful earlier in text the when Fig. 6 is described. - page 9, line 8: "..negative $\Delta$BT values dominate because fewer FLIPs

come from thick and cold areas, decreasing the BT of these pixels..". Why "fewer"?
- Page 9, lines 12-25 and Figure 7: for more clarity, it is suggested to superimpose
averaged $\Delta$BT (FLIP) vs optical depth. These simulations are using CED=20 $\mu$m.
Would the difference between the 8.52 $\mu$m channel and the 3 other channels be as
important for a larger CED, for instance 100 $\mu$m? I think that it should be discussed. -
Page 9, line 25: In my opinion, this sentence is a little weird.

Page 12, line 1; " We can also see in Fig. 8 (b)" Are you actually discussing both Fig.
8a and 8b? Please clarify.

Page 12, lines 7-8: "... When the effects on BTs are roughly the same for all channels,
the MAD... impact on retrieved products may be mitigated (not show here) " Please
develop. Are your referring for instance to larger CED? If yes, I think that it should be
shown.

Page 12, line 14 to page 13, line 24: - The total number of pix-
els found in the 4 optical thickness categories is 52131. I was expecting
40000+10000+1600+400+100+40+16+1= 52157, which is close. Please explain the
difference between these 2 numbers. - The total number of pixels found in the 4
optical thickness heterogeneity parameters categories is 12129. I was expecting
10000+1600+400+100+40+16+1= 12157, which is close. Please explain. How is the
heterogeneity parameter computed? Is the definition given page 13 line 4 the same as
page 5, line 16? I am not sure because the reference is different. Please clarify.

Page 14, lines 11-13: I don't fully understand. Looking at Fig.12, I would say that the
saturation in BT appears at about 8 at 30 degrees and at about 9 at 0 degrees. Please
clarify and perhaps illustrate the "saturation" in Fig.12.

Page 14, line 22: " ..We can also see this in Fig. 13 (f) where.." Please describe Fig.13
first. Fig. 13 and Fig. 12 could actually be shown and discussed before Fig. 11.

6)Technical comments:

Page 1, line 18: in Earth's climate and radiative budget

Page 2, line 1: "cirrus clouds reflect part of the incident solar radiation into space due, but this albedo effect is generally negligible. . ." It looks like something is missing

Page 2, lines 5 and 6: "by taking accurate observations of their optical properties" Please rephrase.

Page 2, line 8: "from microwave to visible ranges" Please specify, for instance spectral ranges.

Page 2, line 35: Top Of Atmosphere (TOA): not consistent with page 1, line 2.

Page 3, line 6: (under 20 $\mu$m). Please specify. Do you mean CED under 20 $\mu$m?

Page 3, lines 17-18: this sentence should be rephrased.

Page 3, lines 22-24: the long sentence is confusing. As it is, I read that the ice crystal model used in MOD06 is simulated by the 3DCLOUD model.

Page 7, line 23: "we see that 3-D and 1-D BTs, decrease " delete comma

Page 8, line 2: ". . .Fauchez et al. (2012, 2014) have shown. . ."

Page 9, line 4: "highly asymmetric regarding" I don't understand.

Page 9, line 7: " for very largest values.." : for the largest values? Please quantify.

Page 9, line 19: " the emission temperature between large optical thicknesses". I don't understand.

Page 11, line 23: '. . ..rapidelly " rapidly

Page 11, line 24: "..through this is more clearly visible at 500 ". even though?

Page 11, line 32: " the single scattering albedo is about 0.3 larger than the value ". Please rephrase.

[Figure]

Page 12, line 32: '. . .we decided pixels. . .″ Please rephrase

Page 13, line 13: ' in on the figures " Please correct

Page 14, line 2:″ and may be generalize to cirrus with similar patterns..″ Please correct generalized

---

## Author Response (AR1)

**Scale dependence of cirrus horizontal heterogeneity effects on TOA measurements.**
**Part I: MODIS brightness temperatures in the thermal infrared**

[revised manuscript text omitted]

approximation (PPHB Cahalan et al. (1994)). Theses approximations are mostly due to time constraints on 3-D forward radiative calculations, the lack of knowledge about the sub-pixel variability and the 3-D structure of the cloud.

Many studies have been conducted in the solar spectral range to better understand the impact of cloud heterogeneities on cloud products. These studies primarily concern warm clouds such as stratocumulus (Varnai and Marshak (2001); Zinner and Mayer (2006); Kato and Marshak (2009); Zinner et al. (2010); Zhang and Platnick (2011); Zhang et al. (2012)) and show that the sign and amplitude of retrieval errors depend on numerous factors, such as the spatial resolution, wavelength, geometry of observation and cloud morphology. In the TIR and for ice clouds, Hogan and Kew (2005) show that radiative transfer (RT) calculations using IPA can change the mean top of atmosphere (TOA) radiative fluxes by $45\ W.m^{-2}$ in the shortwave and by $15\ W.m^{-2}$ in the longwave. Chen and Liou (2006) show that  the broadband thermal cooling rates are increased by around $10\ \%$ in 3-D RT by comparison to 1-D RT. Concerning IR radiances or Brightness Temperatures (BT), Fauchez et al. (2012, 2014) show that heterogeneity effects can significantly influence cirrus optical property retrievals at the 1 km scale of IIR thermal infrared observations, with potentially more than +10 K on TOA BT for heterogeneous pixels, depending also on the cloud altitude. Fauchez et al. (2015) also show that these TOA BT effects result in an overestimate of the retrieved effective diameter by more than $50\ \%$ for small crystals (CED under $20\ \mu m$) and underestimate the retrieved optical thickness by up to $25\ \%$. These errors could significantly influence the cirrus feedbacks assumed in global atmospheric models.

The impact of cloud horizontal heterogeneity on both, TOA radiation and retrieved products depends on the spatial resolution of the instrument (pixel size) and the cloud type. For example, Davis et al. (1997) show for stratocumulus clouds that the heterogeneity impact on cloud optical thickness retrieved from nadir visible radiometric measurements is at a minimum around a pixel size of 1 or 2 km for a small solar zenith angle ($22.5°$). Higher spatial resolutions enhance the IPA error (IPAE), which increases when the photon mean path (before absorption or cloud escape) is equal to or larger than the spatial resolution. Note that here we refer to the word "photon" in the sense of a Fictive Light Particle (FLIP, Pujol (2015)) for stochastic Monte Carlo simulations. 
[revised manuscript text omitted]
.  To be as realistic as possible we have chosen the properties of our simulated cirrus to be close to the average values observed in different studies (references in Table 1) and set the CED to $20\ \mu m$ as the sensitivity of retrievals in the thermal infrared is often limited to CED below $40\ \mu m$. The chosen cirrus geometry which corresponds to an uncinus structure is also the most common form. Two nuances should be mentioned here: i) as seen in Table 1, most of the cirrus parameters cover a wide range of values which means that our simulated case, while realistic in the average sense, does not represent more extreme situations.  ii) this paper is focused only on horizontal heterogeneities: we assume that the vertical variability  in optical properties is negligible compared to the horizontal variability (see Fauchez et al. (2014, 2015)).

**2.2  Ice crystal optical properties**

In this study, we use the same cirrus optical property  coefficients as in the MOD06 product (Holz et al. (2015),Platnick et al. (2016)), namely the severely roughened aggregate of solid columns  of Yang et al. (2013).  The selection of this particle type instead of another habit (or mixture of habits) is based on the study of Holz et al. (2015), who found that this habit provided better consistency between the IR split-window technique and visible and near-/shortwave-/midwave-infrared (VNIR/SWIR/MWIR) retrieval techniques. We assume a constant crystal effective diameter of $20\ \mu m$ throughout the cirrus cloud. Note that TIR retrieval techniques are often limited to effective diameters between 5 and $50\ \mu m$. The choice of a crystal effective diameter of $20\ \mu m$ falls thus almost in the middle of this range. The optical properties of this ice particle as a function at each MODIS channel are shown Table 2 .

**2.3  Radiative transfer**

Radiative transfer computations are performed with the 3-D Monte Carlo code, 3DMCPOL (Cornet et al. (2010), Fauchez et al. (2012, 2014)). In 3DMCPOL, the atmosphere is divided into 3-D volumes named voxels, with constant horizontal sizes (dx, dy) and a variable vertical size (dz) that depends on the atmospheric and cloud vertical stratification. Inside the cloud, each voxel is described by the cloud  scattering properties: the extinction coefficient $\sigma_e$, the single scattering albedo $\varpi_0$, the phase function of the ice crystals and the temperature $T$.

3DMCPOL uses the local estimate method (LEM; Marchuk et al. (1980); Marshak and Davis (2005); Mayer (2009)), which computes the contribution of emission, scattering or reflection events into the detector direction, attenuated by the medium optical thickness between the place of interaction and the detector (Fauchez et al. (2014)). Atmospheric gaseous absorption is
5    parameterized using a correlated k-distribution (Lacis and Oinas (1991); Kratz (1995)) method combined with the equivalence theorem (Partain et al. (2000); Emde et al. (2011)). The equivalence theorem is used in attaching a vector containing the atmospheric absorption attenuation to the photon package, with the vector dimension being equal to the number of bins in the correlated k-distribution. This allows for considerable savings in computational time.

10    In this study, we performed RT calculations for MODIS channels centered at 8.52 $\mu m$, 11.01$\mu m$, 12.03 $\mu m$ and 13.63$\mu m$ in the TIR range. 3DMCPOL computes directly the radiances which are then converted into brightness temperatures (BT), the quantity more commonly used in thermal infrared applications. Figure 4 shows the result of a 3-D BT computation at 12.03 $\mu m$ wavelength and 50 m horizontal spatial resolution for the "cirrus " scene. For this single wavelength and spatial resolution, 100 billion photon packages are computed in 10 days on the NASA NCCS DISCOVER supercom-
15    puter (see acknowledgments) for an accuracy of 0.5 K. As we will explain in section 4, RT computations are performed for the different thermal infrared channels for 1-D and 3-D, and for different spatial resolutions. This yields a large number of cases and a significant computational burden. For this reason, and because Fauchez et al. (2014) showed that radiative heterogeneity effects are linked, to the first order to the optical thickness heterogeneity regardless how the optical thickness is distributed, we chose to simulate only one cirrus case. Nevertheless, the total number of simulated pixels including all wavelength channels
20    and spatial resolutions is 313,000 for the 1-D simulations and 240,000 for the 3-D simulations. Note there are more 1-D computations because they are performed at various scales from 50 m to 10 km while 3-D computations are only performed at 50 m.

**3    Description of horizontal heterogeneity effects**

Clouds have variabilities at many different scales. However, in retrieval algorithms, for simplicity and computational reasons,
25    the independent column approximation (ICA; Stephens et al. (1991)) is commonly applied; cloud layers are assumed to be vertically and horizontally homogeneous with an infinite horizontal extent (i.e. independent of each other). From the satellite retrieval point of view, the ICA is often named IPA for independent pixel approximation (Cahalan et al. (1994)). Obviously, in reality, the pixel is not homogeneous and the radiative transfer between cloudy columns occurs in 3-D. Comparisons of BT simulated with these two RT approaches (IPA and 3-D) allow us to highlight the cloud heterogeneity effects on BT.

We simulated BT with 3DMCPOL at scales ranging from 50 m to 10 km. For each scale, BT values are computed using the 1-D RT assumption at the averaged COT ($BT_{km}^{1D}$ with "x" the scale and "km" the distance unit) and compared with 3-D radiance simulations at the finest field spatial resolution (50 m), arithmetically averaged to the scale being considered and converted to BT (for simplification reason, we will refer this process as BT averaging). The latter are noted $\overline{BT_{50m}^{3D}}^{xkm}$.

The choice of the native spatial resolution for 3-D computations should be much smaller than the photon mean path (distance traveled before absorption or cloud escape) to account for horizontal radiative transport effects. However, the

5   finer is the spatial resolution, the more pixels can communicate. 50 m is the finer spatial resolution 3DMCPOL can achieve in a reasonable computational time for 10 km domain. Table 3 summarizes the number of scattering and photon mean path computed using Marshak and Davis (2005) (chapter 12) for various optical thicknesses and for channels centered at $8.52\ \mu m$, $11.01 \mu m$, $12.03\ \mu m$ and $13.63 \mu m$. Note the number of scatterings increases with the optical thickness and is almost twice as large at $8.52\ \mu m$. Obviously the photon mean geometrical path decreases with optical thickness (for the same cloud geometry)

10   and is about 3 km at $8.52\ \mu m$ for an optical thickness of 1 and only about 0.5 km for an optical thickness of 10.

[revised manuscript text omitted]

15    (0.15 K) while for the others channels it is slightly negative ($\sim$ - 0.06 K). The reason why $\overline{\Delta BT}$ is positive at $13.36$ $\mu m$ and negative for the others channels is due to the larger absorption optical thickness at $13.36$ $\mu m$ causing a HRT effect dominated by the effect described above *(ii)*. Note that, according to MOD06 ice radiative models, the single scattering albedo of large ice crystals in the other channels will converge to values close to that of the $13.36$ $\mu m$ at CED=20 $\mu m$. Therefore, the HRT in the three other channels will be similar to that of the channel centered at $8.52$ $\mu m$.

20    Obviously, the effect of both PPHB and HRT on TOA BT strongly depends on the spatial resolution, as 
[revised manuscript text omitted]

First of all, we can see in Fig. 11 that for off-nadir views, the PPHB is enhanced due to the increasing of the curvature (non linearity) between BT and optical thickness with the view zenith angle. Note that we can also see in this figure that the saturation in BT with respect to changes in optical thickness appears earlier at $\Theta_v = 60°$ ($\tau_{50m}^{1D} \sim 7$) than at $\Theta_v = 30°$ ($\tau_{50m}^{1D} \sim 8$) and $0°$ ($\tau_{50m}^{1D} \sim 9$).

5   Brightness temperatures differences between the viewing geometries can be seen in Fig. 12 with 1-D BT (left column) and 3-D BT (right column). In 1-D we can clearly see that increasing the viewing zenith angle reduce the average brightness temperature and that small differences appears for different azimuth view angles. In 3-D, because the line of sight can cross many different cloudy columns, the radiative field is much more dependent on both the zenith and azimuth view angles. The differences between the 1-D and 3-D fields for oblique views is mostly due to the the THEAB.

[revised manuscript text omitted]
. These results were limited to the channel centered at 11.01 $\mu m$ because computations for other channels were too computationaly expensive. However, optical properties for channels at 11.01 $\mu m$, 12.03 $\mu m$ and 13.36 $\mu m$ are close, leading to similar $MAD(\overline{\Delta BT})$ for nadir view as seen in Fig. 8. $MAD(\overline{\Delta BT})$ for others view angles should be therefore equivalent to the one at 11.01 $\mu m$. Only the 8.52 $\mu m$ channel may have a different behavior. However, considering $MAD(\overline{\Delta BT})$ differences between 11.01 $\mu m$ and 8.52 $\mu m$ in Fig. 8, we can expect that $MAD(\overline{\Delta BT})$ for 8.52 $\mu m$, will be larger for a smaller pixel size due to the larger scattering and the greater horizontal radiative transport.

 **6  Conclusions**

The accurate remote sensing of cirrus clouds is very important in order to improve the parametrization of clouds in climate models and to better understand their role in the Earth-atmosphere system. Cloud heterogeneities may have a significant impact on the accuracy of retrieved cloud optical properties. In this work, we model the impact of cirrus cloud heterogeneities on top-of-atmosphere (TOA) brightness temperatures as a function of the spatial resolution from 50 m to 10 km and at four MODIS thermal infrared channels centered at $8.52\ \mu m$, $11.01\ \mu m$, $12.03\ \mu m$ and $13.36\ \mu m$. A three-dimensional cirrus cloud structure is modeled with the 3DCLOUD cloud generator and radiative transfer simulations are performed with the 3DMCPOL code. In this study,

5   we assume that TOA brightness temperatures differences between BT computation assuming 1-D RT inside a homogeneous pixel and 3-D RT inside a heterogeneous pixel depend on two effects: *(i)* the optical thickness horizontal inhomogeneity leading to the plane parallel approximation bias (PPHB) and the *(ii)* horizontal radiative transport (HRT) effect due to the independent pixel approximation error (IPAE). The cloud vertical heterogeneities of optical properties are here neglected, based on the findings of Fauchez et al. (2014, 2015).

10  As previous studies already showed, the PPHB is the larger heterogeneity effect for nadir observations at the typical spatial resolution of polar orbiters such as AIRS, MODIS or IIR. The PPHB impacts mainly the low spatial resolutions while the IPAE impacts mainly the high spatial resolutions. Although, due to the IPAE, the amplitude of the error in BT can be large at high spatial resolutions, the difference between the errors for different channels is quite small in comparison to the difference at coarse resolution. A similar error between channels can then mitigate the impact on the optical property retrieval. For our

15  simulated cirrus case, we find that the approximate spatial resolution where the PPHB and HRT effects lead to a minimum total effect at nadir is between 100 m and 250 m. In order to extrapolate this result to different cirrus clouds, a sensitivity study has been conducted. The results show that changing the average cloud optical thickness affects the magnitude of the effects but does not significantly change the spatial resolution of the minimum. A space-born radiometer with a nadir spatial resolution between 100 m and 250 m will allow retrieval of cirrus optical properties in the thermal infrared with mitigated

20  overall heterogeneity and radiative effects. In future studies, we will investigate how the errors on COT and CED retrievals due to horizontal inhomogeneities and 3-D effects are scale dependent.

Concerning off-nadir views, when $\Theta_v > 0°$, the line of sight may crosses several different cloudy columns in 3-D RT but not in 1-D RT, leading to the tilted homogeneous extinction assumption bias, THEAB. This increases strongly the mean deviation between 3-D and 1-D BT, especially at fine spatial resolutions. However, in average, an increase in viewing zenith angle

25  decreases the 3-D BT values as well as their heterogeneity, reducing the total error due to PPHB and IPAE. The dependence of the total effect on the azimuth angle could also be important for particular viewing orientations with respect to the cloud. For instance, the cloud heterogeneity, and thus the total effect, is smaller when the line of sight is parallel to the fall streaks of the cloud, and is larger elsewhere. It thus seems that, for arithmetic field average values, the minimum total effect arrises at nadir. Also, the THEAB leads to a shift in the spatial resolution of the minimum total effect toward coarser spatial resolutions.

30  Off-nadir, it is clear that the horizontal and vertical structure of the cloud may change the conclusions. However, we have

chosen the uncinus cirrus structure (with fallstreaks corresponding to intervals of thick and thin optical thicknesses), which is one of the most common among the variety of cirrus structure. We can thus extrapolate that results may be comparable to other uncinus cirrus, but may be different from others structures such as the patchy structures of cirrus floccus.

Note that these simulations where performed for a unique CED of 20 $\mu m$, very common in cirrus clouds but relatively small. However, for example, increasing CED to 80 $\mu m$ leads to a convergence of the single scattering albedo across all the TIR channels towards values between 0.5-0.6 (0.5 being the geometric optics limit). This implies less scattering and thereby horizontal transport in the 8.52 $\mu m$ channel ($\varpi_0 \sim 0.75$ for CED= 20 $\mu m$ in this study). The differences between channels should thus be weaker and consequently the impacts on cloud optical property retrievals, which depend on the radiance relative difference between channels. Also, because single scattering albedo values for all the channels at $D_{eff} = 80\ \mu m$ are close to that at 13.36 $\mu m$ for $D_{eff} = 20\ \mu m$ used in this study, all the channels for $D_{eff} = 80\ \mu m$ will have a similar heterogeneity effect on TOA BT across spatial resolutions than for the 13.36 $\mu m$ channel presented in this study. In Part 2 of this work we will study the impact of cirrus heterogeneities on visible and near infrared MODIS channels and will make comparisons with the result of this present Part 1. We anticipate that the results will be different since 3-D effects are stronger for visible and near infrared wavelength and that solar geometries will play an important role. Additional perspectives will concern the impact of cirrus cloud heterogeneities on the optical property retrievals . Indeed, the dependence of heterogeneity and 3-D effects on the wavelength can be an issue for retrieval techniques using combination of many wavelength ranges (such as optimal estimation methods). Others clouds, such as cumulus or stratocumulus should also be considered, because results are expected to be strongly dependent on the cloud type.

**7   Code availability**

The simulated data used in this study were generated by the 3DCLOUD (Szczap et al. (2014)) and 3DMCPOL (Cornet et al. (2010), Fauchez et al. (2014)) closed-source codes. Please contact the authors for more informations.

*Acknowledgements.* The authors acknowledge the Universities Space Research Association (USRA) through the NASA Postdoctoral Program (NPP) for their financial support.

We thank Dr. Zhibo Zhang and the UMBC High Performance Computing Facility (HPCF) for the use of their computational resources (MAYA). The facility is supported by the U.S. National Science Foundation through the MRI program (grant nos. CNS-0821258 and CNS-1228778) and the SCREMS program (grant no. DMS-0821311), with additional substantial support from the University of Maryland, Baltimore County (UMBC). See www.umbc.edu/hpcf for more information on HPCF and the projects using its resources.

We also thanks the NASA Center for Climate Simulation (NCCS) for the use of their computational resources (Discover). We also deeply acknowledge the two anonymous reviewers whose have contributed with their very relevant comments improving the quality of the manuscript.

[revised manuscript text omitted]

**Figure 7.**  The contribution of the photon horizontal transport to TOA brightness temperature differences between 3-D and 1-D RT at 50m ($\Delta BT = BT^{3D}_{50m} - BT^{1D}_{50m}$) seen from nadir as a function of the optical thickness at 12.03 m (bottom frame). The fraction of pixel for each $\Delta BT$ is shown in the top frame. Positive and negative differences are in red and blue, respectively, at 8.52 $\mu m$ (a), 11.01 $\mu m$ (b), 12.03 $\mu m$ (c) and 13.36 $\mu m$ (d). For these four channels, the $\Delta BT$ percentage of positive values are, 33%, 40%, 41%, 53%, respectively.

[revised manuscript text omitted]

**Table 4.** Ranges of spatial resolutions from which PPHB is larger than mean deviation due to HRT (|HRT|), |HRT| larger than PPHB and where the total absolute difference is minimum (MAD min), for different range of optical thickness $\tau$ and heterogeneity parameter $\rho_\tau$ for four MODIS TIR channels. $(\overline{\rho_\tau} = 1.0)$ and $(\overline{\tau} = 1.4)$ are the average values on the whole cirrus corresponding to the different ranges of $\tau$ and $\rho_\tau$, respectively. See also Fig. 9 and Fig. 10.

| Range | | $\tau < 2.0$ | $2.0 \leq \tau < 3.0$ | $3.0 \leq \tau < 6.0$ | $\tau \geq 6.0$ | $\rho_\tau < 0.3$ | $0.3 \leq \rho_\tau < 0.7$ | $0.7 \leq \rho_\tau < 1.1$ | $\rho_\tau \geq 1.1$ |
|---|---|---|---|---|---|---|---|---|---|
| Effect | Channel | | | $(\overline{\rho_\tau} = 1.0)$ | | | | $(\overline{\tau} = 1.4)$ | |
| | | | | Spatial resolution range | | | | | |
| PPHB>|HRT| | 8.52 $\mu m$ | $\geq 2.5km$ | $\geq 250m$ | $\geq 100m$ | $\geq 100m$ | $\geq 250m$ | $\geq 1km$ | $\geq 250m$ | $\geq 250m$ |
| | 11.01 $\mu m$ | $\geq 500m$ | $\geq 100m$ | $\geq 50m$ | $\geq 50-100m$ | $\geq 250m$ | $\geq 100m$ | $\geq 100m$ | $\geq 100m$ |
| | 12.03 $\mu m$ | $\geq 250m$ | $\geq 100m$ | $\geq 50m$ | $\geq 50-100m$ | $\geq 250m$ | $\geq 100m$ | $\geq 100m$ | $\geq 100m$ |
| | 13.36 $\mu m$ | $\geq 250m$ | $\geq 100m$ | $\geq 50m$ | $\geq 50-100m$ | $\geq 250m$ | $\geq 100m$ | $\geq 100m$ | $\geq 100m$ |
| |HRT|>PPHB | 8.52 $\mu m$ | $\leq 2.5km$ | $\leq 250m$ | $\leq 100m$ | $\leq 100m$ | $\leq 250m$ | $\leq 1km$ | $\leq 250m$ | $\leq 250m$ |
| | 11.01 $\mu m$ | $\leq 500m$ | $\leq 100m$ | $\leq 50m$ | $\leq 50-100m$ | $\leq 250m$ | $\leq 100m$ | $\leq 100m$ | $\leq 100m$ |
| | 12.03 $\mu m$ | $\leq 250m$ | $\leq 100m$ | $\leq 50m$ | $\leq 50-100m$ | $\leq 250m$ | $\leq 100m$ | $\leq 100m$ | $\leq 100m$ |
| | 13.36 $\mu m$ | $\leq 250m$ | $\leq 100m$ | $\leq 50m$ | $\leq 50-100m$ | $\leq 250m$ | $\leq 100m$ | $\leq 100m$ | $\leq 100m$ |
| MAD min | 8.52 $\mu m$ | $\sim 2.5km$ | $\sim 250m$ | $\sim 100m$ | $\sim 100m$ | $\sim 250m$ | $\sim 500m$ | $\sim 250m$ | $\sim 250m$ |
| | 11.01 $\mu m$ | $\sim 500m$ | $\sim 100m$ | $\leq 50m$ | $\leq 50m$ | $\sim 250m$ | $\leq 100m$ | $\leq 100m$ | $\leq 100m$ |
| | 12.03 $\mu m$ | $\sim 250m$ | $\sim 100m$ | $\leq 50m$ | $\leq 50m$ | $\sim 250m$ | $\leq 100m$ | $\leq 100m$ | $\leq 100m$ |
| | 13.36 $\mu m$ | $\sim 250m$ | $\sim 100m$ | $\leq 50m$ | $\leq 50m$ | $\sim 100m$ | $\leq 100m$ | $\leq 100m$ | $\leq 100m$ |

**Reply to Anonymous Referee #1**

We are grateful to referee #1 for carefully reading the manuscript and providing many helpful suggestions.

General comments

The paper compares 3D and 1D Monte Carlo simulations of a cirrus cloud field at four different MODIS wavelength channels in the thermal spectral range. The aim of the study is to investigate the difference in brightness temperature between 1D and 3D radiative transfer in an inhomogeneous cloud field from a nadir satellite perspective and to find the optimal horizontal resolution where the error between the realistic 3D radiative transfer and the commonly used 1D approximations are at a minimum.

Simulations of different horizontal resolutions (50m to 10km) have been performed and differences due to horizontal transport of radiation and the averaging/aggregation of high resolution pixels to coarser resolution, the plane parallel bias, have been addressed.

It was shown that the optimal horizontal resolution varies between 100 and 250m, depending on the wavelength channel. Even at this optimal resolution the difference in brightness temperatures between the 1D and 3D radiative transfer simulation can be up to 7K.

Additionally, sensitivity tests for varying optical properties have been performed. The off-nadir perspective was addressed by simulating one of the four MODIS channels of this study.

With this study, the authors extend former work in this field by showing the difference between 1D and 3D RT brightness temperatures at different horizontal resolutions.

The paper is suitable for publication after minor revision.

General Comments:

The optical thickness used in the paper is not always defined. In one figure, the optical thickness at 0.86m is shown, while most of the manuscript refers to the 12.03m optical thickness. It is not always mentioned which optical thickness is used for the comparisons. The authors might clarify which optical thickness is used where in the study. I would recommend using a single one. What is the reason for choosing that specific wavelength optical thickness?

This is an error in our labeling. Because part 2 of this study is dedicated to visible and near infrared wavelengths, we have kept the same labeling. But in the figure 5, the values are at 12.03 μm. We corrected the label and the caption.

The authors refer often to the mean path of a photon/FLIP when effects of the horizontal resolution are concerned. It might help readers to have a certain number associated with the mean path at the four different wavelengths considered in the work. Maybe the value of the mean path at a certain optical thickness (e.g. 1 or 1.4 as this seems to be the mean optical thickness of the cirrus cloud in this study) could be added.

Following the definition of the mean horizontal displacement given in Marshak and Davis (2005, chapter 12), for a homogeneous cloud, an optical thickness of 1, and wavelengths of 8.02µm, 11.01 µm, 12.03 µm and 13.36 µm we get an approximate mean horizontal transport of 3.34 km, 2.93 km,  2.68  km and   2.59 km, respectively.

Therefore, the mean horizontal displacement is larger than the pixel field of view, especially for 8µm radiances, leading to a stronger effect as seen in Figure 8.   We have added this paragraph to page 7 after line 20: "Table 3 summarizes the number of scattering and photon  mean path computed using Marshak and Davis, 2005 (chapter 12) for various optical thicknesses and for channels centered at 8.52 µm, 11.01 µm , 12.03 µm  and 13.63 µm. Note the number of scatterings increases with optical thickness and is almost twice as large at 8.52 µm than at the other wavelengths. Obviously, the photon mean geometric path decreases with optical thickness (for the same cloud geometry) and is of about 3 km at 8.52 µm for an optical thickness of 1 and only about 0.5 km for an optical thickness of 10."

Some of the figures are hard to read. Especially the choice of red and pink in many of the line plots make it difficult to see the difference in the results. Please see the more specific comments below.

We agree that pink and red lines are difficult to discern especially when the plot is dense. We thus modified the color choice in the figures to improve the clarity. Pink was systematically changed to green.

Many abbreviations are introduced in the introduction. Sometimes the authors use capital letters to show the origin of the abbreviation, but not throughout the text. I recommend doing this throughout the text.

The first letters used for the abbreviation are capitalized only when this is a name (for instance MODIS).

How much different would results of a simulation of the 8.52m channel in the off nadir perspective be? As this channel has a stronger scattering, one might expect stronger 3D effects? I understand that these simulations are expensive, but it might be worth adding this channel to the analysis, or discuss possible differences in the results.

This is indeed an interesting question to assess but unfortunately, as we wrote page 14 lines 7 and 8, the required computational time to perform new off-nadir simulations is too

large. But, regarding the nadir results and differences between 8.52 μm and others channels we can anticipate the results as described below.

We moved line 7-8:"Computations for other channels were too computationally expensive and so a selection of a unique channel was preferred in order to highlight general behaviors related to off-nadir viewing geometries." to the end of the section and added the following paragraph:

"These results were limited to the channel centered at 11.04 μm because computations for other channels were too computationally expensive. However, optical properties for channels at 11.01 μm, 12.03 μm and 13.36 μm are close, leading to similar $MAD(\overline{\Delta BT})$ for nadir view as seen in Fig. 8. $MAD(\overline{\Delta BT})$ for other view angles should therefore be equivalent to the one at 11.04 μm. Only the 8.52 μm channel may have a different behavior. However, considering $MAD(\overline{\Delta BT})$ differences between 11.04 μm and 8.52 μm in Fig. 8, we can expect that $MAD(\overline{\Delta BT})$ for 8.52 μm will be larger for a smaller pixel size due to the larger scattering and the greater horizontal radiative transport."

By the way, we corrected label in Fig. 11 where $\Theta_v$ was inserted instead of $\Phi_v$ for angles of 90 and 180°.

The 'Conclusion' in its current form is a summary of the shown work. An outlook and some discussion about the implications of the results is wanting. Please see the more specific comment below.

Specific comments

**1) Page 2, Line 1: Delete "due"**

Thank you for having seen this typo. We removed it.

**2) Page 2, Line 6: Change "of their optical properties" to "cirrus cloud optical properties"**

Done

**3) Page 2, Line 15-18: This part is challenging to read and understand. I guess that the authors want to point out that the thermal infrared spectral range should (next to the retrieval of temperature/pressure and altitude) also be used for the retrieval of optical properties such as COD and CED? This is part of the motivation for the study and should be pointed out more clearly.**

Yes indeed, we want to point out that several studies have shown the importance of thermal infrared channels for cirrus optical property retrieval. We have reformulated this sentence to: "cirrus optical properties may be retrieved with a better accuracy using a combination of TIR channels instead of VNIR channels, as long as the cirrus is optically thin

enough (with a visible optical thickness between roughly 0.5 and 3) and the CED smaller than 40 µm"

**4) Page 2, Line 19: Comma is missing: "AVHRR, "**

Done

**5) Page 2, Line 20: delete brackets: ((Garnier et al., 2012, 2013))**

Done

**6) Page 2, Line 21: example concerning the capital letters mentioned above: "Optimal Estimation Method" (OEM)**

Because Optimal Estimation Method is not a proper noun such as MODIS or AVHRR etc., we do not believe we should capitalize the first letter of "Optimal Estimation Method".

**7) Page 2, Line 28: "etc.": The authors might add additional reasons or change the sentence to: "due to time constraints on 3-D forward radiative calculations and the lack of: : "**

We have modified the sentence as follow: "3-D forward radiative calculations, the lack of knowledge about the sub-pixel variability and the 3-D structure of the cloud"

**8) Page 3, Line 1: Is longwave here the same as thermal IR?**

In this study, longwave indeed includes thermal infrared but includes longer wavelengths into the infrared spectra.

**9) Page 3, Line 1: Is the cooling rate in 1D too high or too low by 10%?**

We have reformulated this sentence: "the broadband thermal cooling rates are increased by around 10% in 3-D RT by comparison to 1-D RT."

**10) Page 3, Line 13: delete PPHB; it is already introduced at this point.**
Done

**11) Page 5, Line 27: optical thickness at which wavelength?**

At 12.03 um as notified line 29.

**12) Page 6, Line 10: Delete sentence "Note that TIR retrieval techniques are often limited to effective diameters between 5 and 50m." either here or in line 5/6 above.**

We delete it in line 5/6.

**13) Page 6, Line 30: "cirrus 1" - There is only one cirrus case used in this study. I recommend deleting "cirrus 1" in the whole manuscript. Otherwise one would expect more than one scene.**

We agree. We have deleted the "1".

**14) Page 7, Line 18/19: The authors might mention the FLIP mean path as a second motivation for the 50m resolutions already at this point. I saw that it is mentioned later in the text, but it would already be worthy here.**

Actually, the 50m spatial resolution is much finer than the mean horizontal displacement (see earlier comment). As mentioned, we were limited to 50m for computational time reason, but ideally, we would like to simulate up to 10m spatial resolution. At this spatial resolution, a much larger number of pixel can communicate through horizontal radiative transport.

We replaced lines 18/19 by: … "The choice of the native spatial resolution for 3-D computations should be much smaller than the photon mean path (distance travel before absorption or cloud escape) to account for horizontal radiative transport effects. However, 50 m is the finest spatial resolution that 3DMCPOL can achieve in a reasonable computational time for a 10 km domain."

**15) Page 7, Line 21: Mention the wavelength of the optical thickness here. From Figure I take that it is at 0.86m. Why? If the optical thickness is taken in the visible, the 550nm is a common wavelength to use. For the rest of the paper the authors use the 12.03m optical thickness. I suggest using the 12.03m here as well. Additionally, why is the 12.03m wavelength chosen? It is one of the channels of course, but how strong does the optical thickness vary for the wavelength of the other channels?**

The optical thickness is at 12.03 μm, as mentioned earlier, and modified the figure accordingly. A wavelength around 12 μm is typically used as the reference channel in most studies concerning retrieval of cloud properties in the thermal infrared (Garnier et al., 2012, 2013, etc.). Since the extinction coefficients are quite similar between the thermal infrared channels (see Table 2), the difference between optical thickness defined at one channel or another does not have a significant impact.

**16) Page 8, Line 26-28: Something about this paragraph is confusing and requires a better explanation. After reading it several times, I still cannot understand it in full. You point out that extreme values of the BT are smoothed out by the HRT effect. Therefore the difference between 1D and 3D BT should be smaller. As there is more scattering in channel 8.52m, one would expect smaller differences between 1D and 3D BT from the first conclusion. However, Figure 6 and your text show the opposite. This paragraph needs clarification. In addition, the choice of colors, the thickness of the lines and the scale of the y-axis makes this figure hard to read.**

HRT makes the differences between 3-D and 1-D BT not smaller but higher. As mentioned, in 3-D, small BT values (associated with large optical thicknesses) are increased by the HRT and conversely, large BT values are decreased, resulting in a smoothing of the radiative field. Consequently, a 1-D radiative field (where no smoothing occurs) is always more heterogeneous than a 3-D field. Because the smoothing is stronger at 8.52 μm, the difference between 3-D (smooth) and 1-D (unsmooth) BT are larger for this wavelength.

We modified the sentence "This effect is stronger at 8.52 m, where the cloud scattering is significantly larger and cloud absorption smaller. As a result the BT differences between 3-D and 1D are larger at 8.52 μm than at 13.36 μm " to the following:

"The 3-D BT field looks more homogeneous than the 1-D BT field where no smoothing occurs. Because this difference is amplified with the number of scatterings, the differences between 3-D and 1-D for the channel at 8.52 μm are stronger than at 13.36 μm,…"

As previously mentioned, we have converted pink color into green to better contrast with the red in all the figures of the manuscript. We have also increased the linewidth for figure 6.

**17) Page 9, Line 13: replace "smaller scattering" by "less scattering"**
Done

**18) Page 9, Line 19: typo: quite instead of quitte**
Done

**19) Page 11, Line 21: delete "(FLIP average distance before absorption or before leaving the cloud)" - this is explained a few times already**
Done

**20) Page 11, Line 22: typo: rapidly instead of rapidelly**
Done

**21) Page 12, Line 4: typo: Nevertheless instead of Netherless**
Done

**22) Page 12, Line 32: optical thickness at which wavelength?**
At 12.03 μm. We now mentioned that in Page 12, Line 32.

**23) Page 13, Line 12: "we chose to not show" – replace by "we chose not to show"**
Done

**24) Page 14, Line 7: I fully understand that Monte Carlo simulations are very expensive in terms of computational time. However, as scattering is stronger in the 8.52m channel and more horizontal transport of FLIPs between the column should occur, it might be worth adding this channel to the analysis? What result would be expected for the 8.52m channel?**

Unfortunately, it would take too much time to add this channel to the analysis. So we are not able to do it. However, following others results of the paper, we were able to extrapolate the results as answered to your general comments above.

**25) Page 14, Lines 20-22: Reformulate this sentence "In contrast, some lines of sight cross through small optical thicknesses... "**

We rephrased it as: "In contrast, some lines of sight cross small optical thickness where photons emitted from the surface, warmer than the cloud, contribute to the TOA BT"

**26) Page 14, Line 28: Remove "about"**
Done

**27) Page 15, Line 5: typo: lige**
Done

**28) Page 15, Line 24-26: Reformulate sentence: "In this study, we consider..."**

We rephrased it as: "we assume that TOA brightness temperatures differences between computations assuming 1-D RT inside a homogeneous pixel and 3-D RT inside a heterogeneous pixel depend on two effects:"

**29) Conclusion: An outlook concerning the presented work would be beneficial for this section. The authors briefly state what will be shown in a Part 2 paper, however different wavelength channels are involved there. As the choice of the cloud scene seems to have a larger impact on the off-nadir results, additional simulations (in future work) including different cirrus cloud fields might be one aspect. In addition, some discussion about the implications of the results for current cirrus cloud retrievals is wanting. How much would a satellite instrument with a resolution of 100-250m improve current retrievals? One might discuss that in the context of earlier studies (e.g. Fauchez et al., 2015) where the BT differences of 10K was related to ice crystal diameter and retrieved optical thickness. Is there a guess how much this improved resolution, with the following smaller differences in BT would improve the retrieval results? Currently, the conclusion section does not really show any conclusions. It only summarizes the presented work.**

Thank you for this remark, indeed, the conclusion needed more details and perspectives. The conclusion has been significantly modified in new version of the manuscript.

**30) Figure 1, Caption: Delete "cirrus 1" and add "of the study"**

Done

**31) Figure 2: Is the potential temperature and the equivalent potential temperature really the same?**

No, they are different. The equivalent potential temperature is the temperature a parcel of air would reach if all the water would condensate while the potential temperature is the temperature a parcel of air would reach if adiabatically brought to a standard pressure of 1bar.

**32) Figure 3: Which optical thickness is shown in the figure?**
We added 12.03 μm.

**33) Figure 4: Delete "cirrus 1"**
Done

**34) Figure 5: Why do you use the optical thickness at 0.86m here and 12.03m in the following? The colors and especially the markers are hard to separate in this figure. One really has to zoom into the pdf.**
We have corrected the label error. Now 12.03 μm is shown.

**35) Figure 6: The difference between the lines is hard to see, especially the red and pink colors are hard to differentiate. Also, the scale of the y-axis makes it difficult to see the differences properly. The authors might also consider plotting thicker lines.**

We have increased the line thickness and convert the pink lines into green lines to contrast better with the red.

**36) Figure 7: The first sentence of the caption is challenging to understand.**

We have modified this sentence as: "The contribution of photon horizontal transport to TOA brightness temperature differences between 3-D and 1-D RT at 50m ($\Delta BT = BT_{50m}^{3D} - BT_{50m}^{1D}$) seen from nadir as a function of the optical thickness at 12.03 μm (bottom axis). The proportion of pixel relative to each $\Delta BT$ is shown in the top axis."

**37) Figure 8 and following: Please use a different color for the pink lines. Maybe green or orange?**

We are converted pink to green

**38) Figure 9: delete the "t" after "to" at the end of he third line.**
Done

**39) Figure 10: The lines in the upper row are hard to separate. I can see that you want to keep the values of the y-axis constant, but you might think of reducing it to 8 instead of 10? Maybe this would already help?**

Yes, indeed the lines are very closed in this plot we have now rescale the y-axis up to 8.

**40) Table 1: remove "cirrus 1"**

Done

**Technical corrections**
**Please see the "Specific Comments" section.**

**Reply to Anonymous Referee #2**

We would like to thank reviewer #2 very helpful comments who has widely contributed to improve the substance and the form of the paper.

**The authors present the impact of the horizontal heterogeneity of cirrus clouds on TOA brightness temperatures for 4 TIR MODIS channels. The study is based on a "realistic" cirrus case simulated using the 3DCLOUD model, MODIS Collection 6 ice crystal properties, and the 3DMCPOL radiative transfer code. This study discusses the impact of the plane parallel homogeneous bias (PPHB) and of the horizontal radiative transport (HRT) in various conditions of optical depth, optical depth inhomogeneity, and viewing angles. The paper also discusses the optimum horizontal resolution that minimizes the horizontal heterogeneity effects on TOA brightness temperature.**

**General comments:**
**The simulations and the results are solid. The simulated cirrus case is well adapted to illustrate the PPHB and the HRT. However, the impact of this choice on the conclusions of the paper should be discussed. It would be important to know to what extent these results could be generalized. The main characteristics of the simulated cloud should be given in the abstract (lines 7-9).**

After "A realistic 3-D cirrus field is generated by the 3DCLOUD model" we added : "(average optical thickness of 1.4, cloud top and base altitudes at 10 and 12 km, respectively, consisting of aggregate column crystals of $D_{eff}$=20 μm )"

**The reasoning and the story are sometimes difficult to follow. Introductory and linking sentences would be sometimes helpful for the clarity of the manuscript.**

With comments of reviewer #1 and #2 we have improved the clarity of the manuscript, especially in the conclusion.

**My recommendation is to publish this manuscript after clarification on the several points listed above and hereafter.**

**1)Title:**
**The title could specify that this paper discusses cirrus heterogeneity effects on TOA brightness temperatures. "cirrus heterogeneity effects" is too vague, in my opinion.**

We agree that the title is not sufficiently explicit. However, because we would like the first sentence of the title to be the same in part II of this study, we prefer not to mention brightness temperatures at this point. We rephrased the title as follow:
"Scale dependence of cirrus horizontal heterogeneity on TOA measurements. Part I: MODIS brightness temperatures in the thermal infrared channels."

**2)Goal of the paper:**

**Page 3, lines 17 to 21: Please explain the choice of these 4 TIR channels. In which MODIS algorithm(s) are they used and what are the retrieved geophysical parameters?**

These channels are not currently used to retrieve optical properties with MODO6. They are only used by the operational algorithm to infer cloud and surface temperatures. However, as they correspond to atmospheric windows, future versions of the MODIS standard product may include them. This is already the case for instance with the Imaging Infrared Radiometer (IIR; Garnier et al., 2012, 2013) in retrieving optical thickness and particle effective size. At this point of the introduction the utility of these channels has already been discussed (second paragraph). Therefore, we added the following sentence explaining the interest of these channels to the paragraph concerning thermal infrared retrieval technique:

"For example the Split Window Technique (Inoue, 1985) applied to the Advanced Very High Resolution Radiometer (AVHRR Parol et al. (1991)) and the Imaging Infrared Radiometer (IIR) onboard CALIPSO (Garnier et al., 2012, 2013) is used to retrieve CED and COT from the brightness temperature difference of two different window channels *in the infrared atmospheric windows where gaseous absorption is small*."

**".... the impact of horizontal heterogeneity..." Please specify impact on which quantity (TOA BT, optical depth, CED, other?).**

The impact on both, TOA radiation and retrieved product. We mention that after: *"... the impact of horizontal heterogeneity on both, TOA radiation and retrieved products,"*

**3)Realistic cirrus case:**
**The rationale for the choice of the "realistic" cirrus case should be clearly presented. Table 1 should be presented and discussed in more detail. I agree that assuming a "constant" CED of 20 m (page 6, lines 9-12) is "realistic", but it is not typical nor statistically representative. The fact that TIR techniques are often limited to CED between 5 and 50 m (page 6, line 10) clearly does not mean that all CED are so small (as shown in Table 1). Please clarify the rationale.**

We agree that the use of "realistic" needs more details. We have changed the sentence in page 5 lines 32-33: "The simulated cirrus field is thus suitable to study the impact of cloud heterogeneity on radiative transfer at various scales." to: "To be as realistic as possible, we have chosen the properties of our simulated cirrus to be close to average values observed in different studies (reference in Table 1) and set the CED to 20µm as the sensitivity of retrievals in the thermal infrared is often limited to CED below 40 µm. The chosen cirrus geometry, which corresponds to an uncinus structure is  also the most common form. among the variety of cirrus."

We also have added two nuances on the realism of our simulations (after the previous sentence):

"Two nuances should be mentioned here: i) as seen in Table 1, most of the cirrus parameters cover a wide range of values which means that our simulated case, while realistic in the average sense, does not represent more extreme situations. ii) this paper is focused only on…."

**The impact of this choice on the conclusions of the paper should be discussed. In particular, how does it impact the highlighted difference between the 8.52 m channel and the three other channels?**

This is a very interesting remark, indeed when the crystal effective size increase, the single scattering albedo in the different thermal infrared channels tends to converge between 0.5-0.6 (represents the well-known geometric optics lower limit). For instance here are the values for $D_{eff}$= 20 μm: 0.75, 0.42, 0.47 and 0.51 and $D_{eff}$= 80 μm : 0.57, 0.51, 0.53 and 0.53 for channels centered at 8.52, 11.01, 12.03 and 13.36 μm, respectively, used in MOD06.
As you can see, for large crystal size there are less differences between channels which have single scattering coefficients close to the value at 13.36 μm for $D_{eff}$= 20 μm, where the absorption is strong and the scattering weak.

We have added in the conclusion: "Note that these simulations were performed for a unique CED of 20 μm, common in cirrus clouds but relatively small. However, for example, increasing CED to 80 μm leads to a convergence of the single scattering albedo across all TIR channels towards values between 0.5-0.6 (0.5 being the geometric optics limit). This implies less scattering and thereby less horizontal transport in the 8.52 μm channel ($\varpi_0 \approx$ 0.75 in this study). The differences between channels should thus be weaker and consequently the impacts on cloud optical property retrievals, which depend on the radiance relative difference between channels. Also, because single scattering albedo values for all the channels at $D_{eff}$= 80 μm are close to that at 13.36 μm for $D_{eff}$= 20 μm used in this study, all the channels for $D_{eff}$= 80 μm will have a similar heterogeneity effect on TOA BT across spatial resolutions than for the 13.36 μm channel presented in this study."

**Page1, line 7: "A unique but realistic cirrus case is simulated…": Why is the cirrus case "unique"? Do you mean that only one case is simulated?**

We made several simulations from a single cirrus fields. We have rephrased the sentence as "A single but …"

**4)Averaging and aggregation:**
**Please define "averaging" and "aggregation", and use consistent terms throughout the paper. Below are some examples (there are more in the text):**

We should use averaging instead of aggregation, because this is a linear averaging that we performed on BT or optical thickness. We have modified aggregation into averaging in the whole manuscript.

**Page 7: line 17: "…averaged to the scale being considered…". Please detail the averaging process. Which parameter?**

We now mention that this is an arithmetic averaging.

We have changed "RT" into "radiances" which is the quantity arithmetically averaged and then converted to BT. We added: … "averaged to the scale being considered and converted to BT (for simplification reason, we will refer this process as BT averaging."

**Page 7, line 26: "..aggregation.." Please explain what "aggregation" means.**

Aggregation has been replaced by averaging in all the manuscript.

**Page 7, line 30 : "..the averaged BT.." Are you averaging BT? I am surprised because the observations are radiances (same comment page 10, line 8).**

As mentioned two questions earlier we now specify that these are the radiances which are averaged and then converted into BT.

**Page 10, line 7: ", while 1-D BTs are directly computed at the xkm scale after aggregating the 50 m optical thickness" My understanding is that 1-D BT are computed using an averaged optical depth. Is is what you mean?**

Yes this is what we mean.

We rephrase it as: while 1-D BTs are directly computed at the xkm scale from the averaged optical thickness.

**5)Other comments (mostly for clarification):**
**Page 3, lines 24-25: '"we describe the heterogeneity and 3-D effects" For more clarity, it is suggested to specify PPHB and IPAE (or horizontal radiative transport).**

Done

**Page 5, line 9: Figure 1, caption: what is 'Cirrus 1"?**

We have deleted all the reference "cirrus 1" as only one cirrus has been used in this study.

**Page 5, line 29: "For the cirrus used in this study…" Is it cirrus 1 listed in Table 1? Please clarify. Introduce Table 1 earlier. The references listed in Table 1 should be presented and discussed in the text.**

Yes, as only one cirrus has been simulated we removed "cirrus 1" from the text.

Thanks to one of your previous questions, we now give more details in the text concerning this table. We also now reference the authors listed in the caption of the table directly in the text after: "… listed in the literature (…)"

**Page 5, line 34: '….vertical variability of the geometrical and optical thickness.." Please clarify. I don't understand the notion of vertical variability of such quantities.**

We have changed "vertical variability of the geometrical and optical thickness" to "vertical variability in optical properties"

**Page 6, line 3: for more clarity, title of Sect. 2.2 could be "ice crystal optical properties".**

We agree. Done.

**Page 6, line 4: "cirrus optical property parametrization": not entirely clear to me…what about "bulk scattering properties? Is there really a parametrization?**

We have changed "parametrization" to "coefficients". We have also removed "bulk" which is confusing.

**Page 6, lines 5-6: "Note that TIR….between 5 and 50 m". Why this sentence here?**

We deleted this sentence.

**Page 6, lines 7- 9: "…Holz et al. (2015) better consistency between ….the IRsplit-window technique….and (VNIR/SWIR/MWIR) techniques, as well as with lidar retrievals……..". This sentence is very confusing and I do not think that it is entirely correct. You are talking about the consistency between techniques and retrievals. Are you talking about retrieval of optical depth, or CED, or both? "Split-window technique" suggests CED. "Lidar retrievals" suggests "optical depth". Holz et al. (2015) discuss only optical depths, but not CED. Please clarify.**

To avoid confusions, we have remove "lidar retrievals" from sentence.

**Page 6, line 32: "… as will be explained…" Specify in which section.**

We now mention section 4.

**Page 7, line 21: Figure 5 According to the caption, this is now optical depth at 0.86 m not introduced earlier. Please explain.**

This was a labeling error, all optical thicknesses in this study are at 12.03 $\mu$m.

**Page 7, line 33: "decreasing" resolution can be misunderstood. The notion of coarse or fine resolution would avoid any confusion.**

Indeed, we replaced it with "coarsening resolution"

**Page 8, lines 8-13: The authors are discussing Fig. 5, and I am surprised to find these 6 lines with results from another paper. Why not discuss BT 3-D – BT 1D from Fig. 5?**

At this point of the manuscript we do not yet discuss the new results. Thus, we reference previous studies to introduce the new results.

**HRT section: please re-organize the text for more clarity. - Lines 1-2 page 9 (HRT effect only when BT from 3-D and 1-D at the same resolution of 50 m) should be at the beginning of this sub-section, because important for a good understanding of the discussion.**

We believe that this sentence is better here because the assertion "3-D and 1-D BT are computed at the same spatial resolution (50m)" is valid only for Fig. 6 and 7 in this section.

**- Figure 6: it is suggested to add arrows to point to the areas of specific interest discussed in the text. A second panel showing BT differences between 3-D and 1-D could be helpful.**

We think that adding an arrow would not be useful here because we refer in the text to the region as a function of the optical thickness which is clearly seen regarding the right Y-axis. Also, another panel could overload the information in the figure.

**- page 8, line 29: can you give an example of cloud optical property retrievals that use a combination of the 8.52 m and 13.36 m channels?**

The cloud top property retrievals require the use of MODIS channels centered at 8.52 μm and 13.36 μm.

We changed the sentence to "… will impact cloud-top property retrievals (emissivity, cloud top height, etc.)…"

**- Figure 6, caption: I don't see the BTs computed at 11.01 and 12.03 m.**

This was an error, they are not in the figure. We have removed such a reference from the caption.

**Lines 5- 6, page 9 ("as seen in Fig. 6…") could be useful earlier in text the when Fig. 6 is described.**

We modified the sentences "This effect is stronger at 8.52 μm where the cloud scattering is significantly larger and cloud absorption smaller. As a result the BT differences between 3-D and 1D are larger at 8.52 μm than at 13.36 μm " to the following:
"The 3-D BT fields looks more homogeneous than the 1-D BT field where no smoothing occurs. Because this difference is amplified with the number of scatterings, the channel at 8.52 μm shows a stronger smoothing than at 13.36 μm, …"

**- page 9, line 8: "..negative BT values dominate because fewer FLIPs come from thick and cold areas, decreasing the BT of these pixels..". Why "fewer"?**

The "fewer" is confusing and useless, we have removed it.

**- Page 9, lines 12-25 and Figure 7: for more clarity, it is suggested to superimpose averaged BT (FLIP) vs optical depth. These simulations are using CED=20 m.
Would the difference between the 8.52 m channel and the 3 other channels be as important for a larger CED, for instance 100 m? I think that it should be discussed.**

We do not quite understand what is meant by "superimpose averaged BT (FLIP) vs optical thickness".

We added this sentence in page 9 before line 25: "Note that, according to MOD06 ice radiative models, the single scattering albedo of large ice crystals in the other channels will converge to values close to that of the 13.36 μm channel at CED=20 μm. Therefore, the HRT in the three other channels will be similar to that of the channel centered at 13.36 μm. "

**-Page 9, line 25: In my opinion, this sentence is a little weird.**

We have clarified this sentence as follows: "Obviously, the effect of both PPHB and HRT on TOA BT strongly depends on the spatial resolution as discussed in the next section."

**Page 12, line 1; " We can also see in Fig. 8 (b)" Are you actually discussing both Fig. 8a and 8b? Please clarify.**

Yes, we refer at both Fig. 8(a) and (b). We thus removed the "(b)"

**Page 12, lines 7-8: "… When the effects on BTs are roughly the same for all channels, the MAD… impact on retrieved products may be mitigated (not show here) " Please develop. Are your referring for instance to larger CED? If yes, I think that it should be shown.**

No, we just mention here that differences between the curves for small pixel sizes are smaller than for large pixel sizes. This means that the horizontal heterogeneity and 3-D effects are less wavelength dependent for high spatial resolutions than for coarse ones. We added these sentences: "Note that these differences are dependent on the CED for which the single scattering albedo varies with wavelength. For very large CED (>80 μm) the single

scattering varies less between wavelengths (about the value of CED =20 μm for 13.36 μm), reducing $\overline{\Delta BT}$ differences between channels and therefore the overall impact in the retrieval."

**Page 12, line 14 to page 13, line 24: - The total number of pixels found in the 4 optical thickness categories is 52131. I was expecting 40000+10000+1600+400+100+40+16+1= 52157, which is close. Please explain the difference between these 2 numbers. - The total number of pixels found in the 4 optical thickness heterogeneity parameters categories is 12129. I was expecting 10000+1600+400+100+40+16+1= 12157, which is close. Please explain.**

We made a mistake when calculating the number of pixels for the very large optical thicknesses and very large optical thickness heterogeneity. Because of rounding, we missed some pixels. We have corrected the value now to be 1,089 and 117 pixels, respectively.

**How is the heterogeneity parameter computed? Is the definition given page 13 line 4 the same as page 5, line 16? I am not sure because the reference is different. Please clarify.**

This is Szczap et al., (2000) and not (2014), thank you for having notified this.

**Page 14, lines 11-13: I don't fully understand. Looking at Fig.12, I would say that the saturation in BT appears at about 8 at 30 degrees and at about 9 at 0 degrees. Please clarify and perhaps illustrate the "saturation" in Fig.12.**

We agree and have modified the values accordingly.

**Page 14, line 22: " ..We can also see this in Fig. 13 (f) where.." Please describe Fig.13 first. Fig. 13 and Fig. 12 could actually be shown and discussed before Fig. 11.**

We agree with the reviewer. Fig.12 becomes fig. 11, fig. 13 becomes fig. 12, and fig. 11 becomes fig 13. The text associated to the figures has also changed.

**6)Technical comments:**

**Page 1, line 18: in Earth's climate and radiative budget**

Done

**Page 2, line 1: "cirrus clouds reflect part of the incident solar radiation into space due, but this albedo effect is generally negligible..." It looks like something is missing**

The "due" was too much. We have remove it.

**Page 2, lines 5 and 6: "by taking accurate observations of their optical properties"**

**Please rephrase.**

".. by improving the retrieval of cirrus cloud optical properties"

**Page 2, line 8: "from microwave to visible ranges" Please specify, for instance spectral ranges.**

Done (few millimeters) and (up to 0.4 μm)

**Page 2, line 35: Top Of Atmosphere (TOA): not consistent with page 1, line 2.**

We remove the capital letter in page 2 line 35 and add " the"

**Page 3, line 6: (under 20 m). Please specify. Do you mean CED under 20 m?**

Yes, we now specify CED.

**Page 3, lines 17-18: this sentence should be rephrased.**

This sentence was unclear, we rephrased it to: "However, because such studies focus only on stratocumulus clouds, which are very different from cirrus and because they were only conducted for the common imager solar reflectance channels, their conclusions cannot be simply extrapolated."

**Page 3, lines 22-24: the long sentence is confusing. As it is, I read that the ice crystal model used in MOD06 is simulated by the 3DCLOUD model.**

We added a "then we discuss on" between the two parts of the sentence.

**Page 7, line 23: "we see that 3-D and 1-D BTs, decrease " delete comma**

Done

**Page 8, line 2: "...Fauchez et al. (2012, 2014) have shown..."**

Done.

**Page 9, line 4: "highly asymmetric regarding" I don't understand.**

We have replaced it by "very dependent on"

**Page 9, line 7: " for very largest values.." : for the largest values? Please quantify.**

We changed it to "very large values"

**Page 9, line 19: " the emission temperature between large optical thicknesses". I don't understand.**

We replace it by " the brightness temperature.."

**Page 11, line 23: '....rapidelly " rapidly**

Done

**Page 11, line 24: "..through this is more clearly visible at 500 ". even though?**

Yes, we replaced "through" by "even though"

**Page 11, line 32: " the single scattering albedo** is about 0.3 larger **than the value ". Please rephrase.**

We rephrased it: "... 0.3 above the value..."

**Page 12, line 32: '...we decided pixels..." Please rephrase**

We replaced it by "we sampled"

**Page 13, line 13: ' in on the figures " Please correct**

We removed the "on"

**Page 14, line 2:" and may be generalize to cirrus with similar patterns.." Please correct generalized**

Done.